# The effect of casein glycomacropeptide versus free synthetic amino acids for early treatment of phenylketonuria in a mice model

Kirsten K. Ahring[1]*, Frederik Dagnæs-Hansen[2], Annemarie Brüel[2], Mette Christensen[3], Erik Jensen[4], Thomas G. Jensen[2], Mogens Johannsen[5], Karen S. Johansen[2], Allan M. Lund[3], Jesper G. Madsen[2], Karen Brøndum-Nielsen[1], Michael Pedersen[6], Lambert K. Sørensen[5], Mads Kjolby[2,7,8], Lisbeth B. Møller[9]

1 Departments of Paediatrics and Clinical Genetics, PKU Clinic, Kennedy Center, Copenhagen University Hospital, Rigshospitalet, Denmark, 2 Department of Biomedicine, Health, Aarhus University, Aarhus, Denmark, 3 Departments of Paediatrics and Clinical Genetics, Centre for Inherited Metabolic Diseases, Copenhagen University Hospital, Rigshospitalet, Denmark, 4 Arla Foods Ingredients Group P/S, Viby J, Denmark, 5 Department of Forensic Medicine, Aarhus University, Skejby, Aarhus, Denmark, 6 Comparative Medicine Lab, Aarhus University Hospital, Aarhus, Denmark, 7 Steno Diabetes Center Aarhus, Aarhus University Hospital, Aarhus, Denmark, 8 Department of Clinical Pharmacology, Aarhus University Hospital, Aarhus, Denmark, 9 Department of Clinical Genetics, Kennedy Center, Copenhagen University Hospital, Rigshospitalet, Denmark

* kirsten.kiaer.ahring@regionh.dk

**Data Availability Statement:** All relevant data are within the paper.

## Abstract

### Introduction

Management of phenylketonuria (PKU) is mainly achieved through dietary control with limited intake of phenylalanine (Phe) from food, supplemented with low protein (LP) food and a mixture of free synthetic (FS) amino acids (AA) (FSAA). Casein glycomacropeptide (CGMP) is a natural peptide released in whey during cheese making by the action of the enzyme chymosin. Because CGMP in its pure form does not contain Phe, it is nutritionally suitable as a supplement in the diet for PKU when enriched with specific AAs. Lacprodan® CGMP-20 (= CGMP) used in this study contained only trace amounts of Phe due to minor presence of other proteins/peptides.

### Objective

The aims were to address the following questions in a classical PKU mouse model: Study 1, off diet: Can pure CGMP or CGMP supplemented with Large Neutral Amino Acids (LNAA) as a supplement to normal diet significantly lower the content of Phe in the brain compared to a control group on normal diet, and does supplementation of selected LNAA results in significant lower brain Phe level?. Study 2, on diet: Does a combination of CGMP, essential (non-Phe) EAAs and LP diet, provide similar plasma and brain Phe levels, growth and behavioral skills as a formula which alone consist of FSAA, with a similar composition?.

### Material and methods

45 female mice homozygous for the *Pah^enu2^* mutation were treated for 12 weeks in five different groups; G1(N-CGMP), fed on Normal (N) casein diet (75%) in combination with

**Funding:** The research was conducted as part of an Industrial PhD program, which in Denmark is a three-year industrially focused PhD project where a student (Kirsten Ahring) is hired by a company (Arla Foods Ingredients (AFI)) and enrolled at a university (Copenhagen University) at the same time. The industrial PhD program was managed by Innovation Fund Denmark and partly funded by AFI (https://ufm.dk/en/research-and-innovation/funding-programmesfor-research-and-innovation/find-danish-fundingprogrammes/programmes-managed-byinnovation-fund-denmark/industrial-phd). AFI produces and commercialize the food ingredient; Casein glycomacropeptide which was used in the research.

**Competing interests:** Erik Jensen was the industrial supervisor for the above described PhD project, and his mandatory obligation was to contribute with inputs to study design, formulation of test diets and study manuscript according to agreements and ethical standard. However, Erik Jensen did not have the final decision about these issues and this affiliation does not alter our adherence to PLOS ONE policies on sharing data and materials. Erik Jensen has sent a written signed statement regarding this (separate file).

**Abbreviations:** CGMP-20, (product name Lacprodan® CGMP-20); G1(N-CGMP), Normal (N) casein diet (75%) in combination with CGMP (25%); G2 (N-CGMP-LNAA), Normal (N) casein diet (75%) in combination with CGMP (19,7%) and selected LNAA (5,3% Leu, Tyr and Trp); G3 (N), normal casein diet (100%); G4 (CGMP-EAA-LP), CGMP (70,4%) in combination with essential AA (19,6%) and LP diet; G5 (FSAA-LP), FSAA (100%) and LP diet.

CGMP (25%); G2 (N-CGMP-LNAA), fed on Normal (N) casein diet (75%) in combination with CGMP (19,7%) and selected LNAA (5,3% Leu, Tyr and Trp); G3 (N), fed on normal casein diet (100%); G4 (CGMP-EAA-LP), fed on CGMP (70,4%) in combination with essential AA (19,6%) and LP diet; G5 (FSAA-LP), fed on FSAA (100%) and LP diet. The following parameters were measured during the treatment period: Plasma AA profiles including Phe and Tyr, growth, food and water intake and number of teeth cut. At the end of the treatment period, a body scan (fat and lean body mass) and a behavioral test (Barnes Maze) were performed. Finally, the brains were examined for content of Phe, Tyr, Trp, dopamine (DA), 3,4-dihydroxyphenylacetic acid (DOPAC), serotonin (5-HT) and 5-hydroxyindole-acetic acid (5-HIAA), and the bone density and bone mineral content were determined by dual-energy x-ray absorptiometry.

## Results

Study 1: Mice off diet supplemented with CGMP (G1 (N-CGMP)) or supplemented with CGMP in combination with LNAA (G2 (N-CGMP-LNAA)) had significantly lower Phe in plasma and in the brain compared to mice fed only casein (G3 (N)). Extra LNAA (Tyr, Trp and Leu) to CGMP did not have any significant impact on Phe levels in the plasma and brain, but an increase in serotonin was measured in the brain of G2 mice compared to G1. Study 2: PKU mice fed with mixture of CGMP and EAA as supplement to LP diet (G4 (CGMP-EAA-LP)) demonstrated lower plasma-Phe levels but similar brain- Phe levels and growth as mice fed on an almost identical combination of FSAA (G5 (FSAA-LP)).

## Conclusion

CGMP can be a relevant supplement for the treatment of PKU.

## Introduction

Phenylketonuria (PKU) (OMIM 261600) is an inherited, autosomal recessive metabolic disorder, caused by reduced conversion of phenylalanine (Phe) to tyrosine (Tyr), due to deficient phenylalanine hydroxylase (PAH) activity, resulting in increased blood Phe levels [1–4]. Without treatment PKU results in severe mental retardation, microcephaly, epilepsy and other neurological symptoms [5]. Also, decreased bone mineral density (BMD) have been reported in patients with PKU [6]. Lifelong adherence to low protein (LP) diet supplemented with free synthetic (FS) amino acids (AA) (FSAA) is recommended, but compliance is often poor, partly due to the taste of the FSAA mixtures [7–9]. Therefore, new treatment options are explored.

Casein glycomacropeptide (CGMP) is a natural 64-amino acid peptide released in whey during cheese making by the action of the enzyme chymosin [10]. CGMP in its pure form does not contain Phe, and this makes it suitable as supplement for patients with PKU when supplemented with adequate amounts of essential amino acids (EAA), tyrosine (Tyr), tryptophan (Trp), histidine (His), arginine (Arg), methionine (Met), lysine (Lys), and leucine (Leu). CGMP contains 2- to 3-fold of isoleucine (Ile), valine (Val) and threonine (Thr) compared to the concentrations found in other dietary proteins [11]. Lacprodan® CGMP-20 (= CGMP) used in this study contained trace amounts of Phe due to minor presence of other proteins/peptides. Lacprodan® CGMP-20 will here be referred to as CGMP.

In recent years, trials for evaluation of safety, acceptability and efficacy in mice and humans have demonstrated that CGMP, supplemented with EAA to make it nutritionally adequate, is a safe alternative to conventional treatment with FSAA mixtures [12–17].

A large number of studies have demonstrated that LNAAs, and especially Leu, Tyr and Trp have the ability to reduce Phe entering the brain [18–21]. As CGMP contains large amount of Trp, CGMP might potentially hamper Phe entering the brain.

Furthermore, since CGMP is a 64-amino acid natural peptide and absorbed as di- and tripeptides as well as free AAs from the gut and subsequently from blood to the brain, this can potentially lead to different concentrations of AA and metabolites in blood and brain compared to absorption of FSAA.

The aims of this study were to address the following questions: Study 1, off diet: Can pure CGMP or CGMP supplemented with the LNAAs; Tyr, Leu and Trp significantly lower the content of Phe in the brain? Does the supplementation of extra LNAA results in significantly lower brain Phe level compared to supplementation of CGMP alone? Study 2, on diet: Does a combination of CGMP, EAAs and LP diet, provide similar plasma and brain Phe levels, growth and behavioral skills as a formula with a similar combination of pure FSAA?

This study is to our knowledge the first study to compare almost identical combinations of FSAA and CGMP in a PKU mouse model. Furthermore, it is noteworthy, that the length of this study is 12 weeks diet intervention, which is about twice as long as similar studies [12, 22].

## Material and methods

### Animals

The facilities and protocols used in this study were approved by The Danish Experimental Animal Inspectorate (License no. 2013-15-2934-00878). Experimental animals were produced by breeding B6;BTBR mice homozygous for the *Pah*$^{enu2}$ mutation to yield homozygous PKU mice [23]. This mouse strain, the *Pah*$^{enu2}$, was a mix between the black and tan, brachyury (BTBR) mouse and the C57Bl/6J (B6) mouse. Further genotyping of 6 randomly chosen animals from the mouse strain showed that they were between 93%-95% identical with B6 compared to BTBR, which confirms the almost identical behavior with B6 and equally difference with BTBR [22, 24]. The breeding animals were maintained on Phe-free semi synthetic diet (Harlan Laboratories), also during pregnancy and weaning period. As the diet was free of Phe, the drinking water was supplemented with Phe (Sigma-Aldrich Chemie) to a final concentration of 563 mg/l during that period. Mice were housed in open cages in groups of 2–5 mice. Cages were with Tapvei 2HV bedding (L: 37 cm × W: 21 cm × H: 15 cm; a maximum of 5 mice per cage). The facility was temperature-controlled at 22 ˚C on a 12-12-h light-dark cycle, and the mice were fed ad libitum with the different diets and had free access to water. Environmental enrichment included house, Tapvei s-bricks and Ancare NES3600 nestlets (Brogaarden). One mouse (control group G3 (N)) died after week 5. During the experiment all the animals were weighted every week. The protocol can be found here [25].

The mouse strain used in this study was obtained by breeding mice homozygous for the *PAH*$^{enu2}$ mutation, and unfortunately, no wild-type mouse of this strain is available. However, as the purpose of the present study was to investigate the effect of different diets, it is still possible to achieve even without a normal control. The group of mice on normal casein diet G3(N) is used as a control for the different diets used.

### The research diets

The diets were made by Research Diets according to recipes supplied by the investigator, irradiated and stored at 5 ˚C until use. The AA profiles were made to meet the requirements for

mice, which included 5 g/100 g protein equivalent (PE) supplementation of Arg and Met (4.5 g/100 g PE) [26, 27]. All the diets were isocaloric with the AA/protein source being the only source of variation. The diet for group 1 (G1 (N-CGMP)) contained 25% CGMP and 75% casein (Normal diet, N). The diet for group 2 (G2 (N-CGMP-LNAA)) was similar to G1 but supplemented with Trp, Tyr and Leu to test the potential blocking effect in the gut and at the blood brain barrier. The diet for group 4 (G4 (CGMP-EAA-LP)) consisted of CGMP supplemented with EAA to meet the requirements for mice [26, 27]. Group 5 (G5 (FSAA-LP)) was on LP diet supplemented with FSAA and had a similar AA profile as G4 (CGMP-EAA-LP). The diet for group 3 (G3 (N)) is a complete "normal diet (N), consisting of casein (MIPRO-DAN® *30)*, sucrose, corn starch, corn oil, cellulose, mineral- and vitamin-mix. Basically, the nutritional value is the same in all diets. Phe was added to the drinking water in G4 (CGMP-EAA-LP) and G5 (FSAA-LP) (final concentration 563 mg/l) to prevent Phe deficiency since the CGMP and AA mixture served as the sole protein source for these mice. The mice in G1 (N-CGMP), G2 (N-CGMP-LNAA) and G3 (N) drank normal water. The detailed contents of the diets are shown in Tables 1 and 2. Food and water intake were calculated as the mean of intake /week /mice by dividing the total amount of intake in each cage per week (week 1–11), divided by the number of animals per cage.

### Weekly Phe and Tyr analyses

A blood drop was collected on filter paper (DBS) Whatman FTA/FTA Elute cards (Whatman, GE Healthcare Europe) every week (week 1–12), except for week 6 and 12. At week 6 and 12 bloods for total AA profile analysis were taken. Phe and Tyr content in a punch from the filter paper blood sample was analysed by High Performance Liquid Chromatography (HPLC)-Tandem Mass Spectrometry (TMS) at *Statens Serum Institute*, Denmark, according to standard protocols.

### AA profile analysis in plasma

Blood samples were taken for AA profile analysis after 3.5–5.5 hours of fasting by puncture of the mandibular vein at week 6 and 12 during diet treatment. Plasma was isolated by centrifugation at $3000 \times g$ for 10 min at 4 ˚C and stored at -80 ˚C prior to analysis. A small pilot study was performed to validate the test.

Analysis of free AA in plasma was carried out using a Waters Acquity Ultra-Performance Liquid Chromatography (UPLC) (Waters) system with an integrated photodiode array detector and the Mass Trak AA Solution Kit (Waters) according to standard protocols with the slight modifications described by Peake *et al* [29]. All conditions are available upon request.

### Body composition analysis

At the end of the feeding period (week 13), the animals were analysed for body composition by magnetic resonance (MR) scanning (EchoMRI-100H).

### Behavior performance test (Barne's maze behavioral test)

After 1 week of acclimatisation (week 13) a behavioural test (Barne's Maze) was performed to test short term and long-term memory (week 14–15). Behavioural testing were performed between 8 am and 4 pm. The scientist performing the behavioral studies was blinded to the experimental group. The tests were carried out to asses any cognitive impact of the diet upon the test animals' memory and spatial learning abilities [30]. Tests were performed on a circular table with 20 holes spread around the circumference of the table. One of the 20 holes was

**Table 1. Composition of the experimental diet for the 5 groups (G1, G2, G3, G4, G5) and content of AA in CGMP-20 (g AA/100 g protein).**

| | Ingrediens | OFF DIET | | | ON DIET | |
|---|---|---|---|---|---|---|
| | Group | G1 (N-CGMP) | G2 (N-CGMP-LNAA) | G3 (N) | G4 (CGMP-EAA-LP) | G5 (FSAA-LP) |
| | | g | g | g | g | g |
| | Casein (Miprodan® 30) | 155.25 | 155.25 | 207.00 | 0 | 0 |
| | Sucrose | 458.63 | 458.75 | 465.98 | 454.75 | 471.68 |
| | Corn Starch | 150.00 | 150.00 | 150.00 | 150.00 | 150.00 |
| | Corn Oil | 100.00 | 100.00 | 100.00 | 100.00 | 100.00 |
| | Cellulose | 30.00 | 30.00 | 30.00 | 30.00 | 30.00 |
| | Mineral Mix. AIN-76 (170915) | 35.00 | 35.00 | 35.00 | 35.00 | 35.00 |
| | Vitamin Mix. standard | 10.00 | 10.00 | 10.00 | 10.00 | 10.00 |
| | Ethoxyquin. antioxidant | 0.02 | 0.02 | 0.02 | 0.02 | 0.02 |
| | Choline bitartrate | 2.00 | 2.00 | 2.00 | 2.00 | 2.00 |
| **Lacprodan® CGMP-20** g AA/ 100 g protein | | 59.10 | 46.63 | | 153.70 | |
| 6.4 | Ala | | | | | 8.50 |
| 0.3 | Arg | | | | 11.42 | 11.86 |
| 9.2 | Aspartic Acid | | | | | 11.98 |
| 0.08 | Cys | | | | | 0.15 |
| 21.1 | Glutamic Acid | | | | | 26.30 |
| 1.2 | Glycine | | | | | 1.52 |
| 0.2 | His | | | | 4.41 | 4.57 |
| 11.5 | Ile | | | | | 14.38 |
| 2.5 | Leu | | 1.36 | | 15.14 | 18.24 |
| 6.4 | Lys | | | | 3.46 | 11.86 |
| 2 | Met | | | | 5.31 | 8.20 |
| 0.2 | Phe | | | | 0.45 | 0.65 |
| 12.6 | Pro | | | | | 16.31 |
| 8.5 | Ser | | | | | 10.37 |
| 18.1 | Thr | | | | | 23.42 |
| 0.04 | Trp | | 3.43 | | 3.66 | 3.66 |
| 0.06 | Tyr | | 7.56 | | 16.36 | 16.41 |
| 9.5 | Val | | | | 2.20 | 13.85 |
| 109.88 | TOTAL | 1000.00 | 1000.00 | 1000.00 | 1000.00 | 1000.00 |

The composition is based on the following recommendation: WHO Technical Report Series. Protein and amino acid requirements in human nutrition. 2007 [28]. See also Table 2.

* G1-G5 refers to groups 1–5. In addition to CGMP, G2 and G4 received additional AA in form of FSAA (% indicates fraction of AA received in the form of FSAA): Off diet (G2): Leu: 53.75%, Trp: 99.42% and Tyr: 99.6%. On diet (G4): Arg: 96.25%, His: 93.44%, Leu: 79.76%, Lys: 30.42%, Met: 64.60%, Trp: 98.36%, Tyr: 99.44% and Val: 15.56% G4 and G5 receive in addition Phe from the drinking water (0,563mg Phe/ml). Each animal drank on average 30.3 ml (G4 ~17mg Phe/week) and 38.2 ml (G5~ 22mg Phe/week). In comparison each animal ate an average of 14.6g /week (G4) and 15.4g/week (G5) respectively.

designed as an escape hole, allowing the animals to escape the open surface of the table into a container below, thus utilizing the natural instinct of mice to avoid open spaces and seek cover. The remaining holes provided the mice with only shallow cover. Visual clues, consisting of different colors and patterns, were placed on the walls surrounding the table and the relation between the clues and the escape hole remained unchanged throughout the experimental period. The animals were allowed to rest for 1 hour before testing commenced. Each animal was then placed in its own container with bedding and a metal cover. At the beginning of behavioral trials, the animals were placed centrally on the table, to rest under a cover for 1

**Table 2. Analysis of diets G1-G5.** The diets G4 (CGMP-EAA-LP) and G5 (FSAA-LP) receive in addition Phe from the drinking water (0,563mg Phe/ml).

| Ingredients | OFF DIET | | | ON DIET | |
| --- | --- | --- | --- | --- | --- |
| Group | G1 (N-CGMP) | G2 (N-CGMP-LNAA) | G3 (N) | G4 (CGMP-EAA-LP) | G5 (FSAA-LP) |
| Protein (%) | 18.0 | 17.9 | 18.2 | 17.2 | 15.6 |
| Carbohydrate (%) | 63.2 | 63.5 | 64.4 | 64.2 | 68.3 |
| Fat (%) | 9.2 | 9.3 | 9 | 9.1 | 10.1 |
| Minerals (%) | 3.3 | 3.3 | 3.2 | 3.5 | 2.4 |
| Moisture (%) | 6.3 | 6 | 5.3 | 6.1 | 3.6 |
| g AA/100g | | | | | |
| Ala | 0.73 | 0.68 | 0.59 | 0.81 | 1.02 |
| Arg | 0.51 | 0.52 | 0.70 | 1.14 | 1.1 |
| Aspartic Acid | 1.42 | 1.35 | 1.37 | 1.12 | 1.07 |
| Cys | 0.09 | 0.08 | 0.10 | 0.04 | 0.03 |
| Glutamic Acid | 4.05 | 3.86 | 4.23 | 2.6 | 2.59 |
| Glycine | 0.32 | 0.31 | 0.36 | 0.15 | 0.15 |
| His | 0.41 | 0.42 | 0.56 | 0.27 | 0.23 |
| Ile | 1.23 | 1.13 | 0.98 | 1.37 | 1.43 |
| Leu | 1.43 | 1.53 | 1.81 | 1.72 | 1.77 |
| Lys | 1.40 | 1.36 | 1.56 | 1.18 | 0.92 |
| Met | 0.48 | 0.48 | 0.52 | 0.75 | 0.73 |
| Phe | 0.74 | 0.74 | 1.01 | 0.06 | 0.06 |
| Pro | 2.15 | 2.01 | 2.12 | 1.57 | 1.58 |
| Ser | 1.19 | 1.12 | 1.10 | 0.99 | 0.96 |
| Thr | 1.46 | 1.30 | 0.83 | 2.22 | 2.35 |
| Trp | 0.16 | 0.44 | 0.21 | 0.3 | 0.3 |
| Tyr | 0.67 | 1.33 | 0.94 | 1.41 | 1.44 |
| Val | 1.33 | 1.25 | 1.25 | 1.36 | 1.26 |
| TOTAL | 19.77 | 19.91 | 20.24 | 19.06 | 18.99 |
| Phe in water | No | No | No | Yes | Yes |

Each animal drank on average 30.3 ml (G4 ~17mg Phe/week) and 38.2 ml (G5~ 22mg Phe/week). In comparison each animal ate an average of 14.6g /week (G4) and 15.4g/week (G5) respectively.

min. The cover was then removed, and the animal given 3 min to find the escape hole. If the hole was not found within that time, the hole was shown to the animal by the operator. The bottoms of the non-escape holes were randomly switched between trials as well. Animal performance was recorded using AnyMaze (ANY-Maze) and the metrics were: total time on table, distance traveled, mean speed, time spent to locate the escape hole and whether the trial was successful or not, i.e. the animal located the escape hole within 3 min. On day 1 of testing each animal would complete 5 trials and subsequently 4 trials on day 2 to 5. Once the animals had completed these trials, they were given 7 days rest, then put through 4 additional trials on day 11 in order to test long term memory.

## Brain and bone removal

After week 15, the animals were placed under anaesthesia using 4% isoflurane in air and euthanized by cervical dislocation. The brain was harvested, transferred to a petri dish on ice and then dissected into six parts; cerebellum, brain stem, hypothalamus, parietal cortex, anterior piriform, cortex and olfactory bulb and the parts were snap frozen in liquid nitrogen separately and stored at −80 ˚C until analysis. Afterwards, the two femoral bones were dissected out by opening the hip and knee joints, stripped from muscles and soft connective tissue, and frozen in Ringer-lactate in Eppendorf tubes at -20 ˚C until analysis [31].

## Analysis of Phe, Tyr, Trp, neurotransmitters, and their metabolites in the brain regions

Analysis for content of Phe, Tyr, Trp, 5-hydroxyindole-acetic acid (5-HIAA), 3,4-dihydroxy-phenylacetic acid (DOPAC), dopamine (DA) and serotonin (5-HT) was performed at Department of Forensic Medicine, Aarhus University Hospital, Denmark. The brain components (2–140 mg) were homogenised using a Precellys tissue homogeniser (Bertin Technologies) in a volume of 1.1 mL cold 78% acetonitrile containing ascorbic acid and stable isotope labelled internal standards. The AA were analysed directly after dilution of the extract. The other substances were cleaned up by solid phase extraction on ion exchange sorbents. The measurements were performed by (UPLC-MS/MS). The UPLC system was a Waters Acquity system that consisted of a binary pump, a flow-through-needle sample manager thermostated at 5±2 ˚C and a column oven set at 45±2 ˚C (Waters). The MS/MS was a Waters Xevo TQ-S triple-quadrupole instrument with an electrospray ionisation ion (ESI) source. The separation was performed using a reversed-phase HSS T3 column (1.8 μm, 200 Å, 2.1 mm I.D. × 100 mm) (Waters). The mobile phases A and B consisted of 10% methanol and methanol/acetonitrile (1+1), both acidified with formic acid and acetic acid. A 10 μL volume was injected onto the column running 100% mobile phase A. The mobile phase was changed through a linear gradient to 90% A and 10% B over 4 min. Then the column was washed and conditioned before next injection. The primary ion transitions used for quantification and qualification were $m/z$ 137 > 65 and 91 (DA), $m/z$ 160 > 77 and 132 (5-HT), $m/z$ 192 > 118 and 146 (5-HIAA), $m/z$ 123 > 123 and 95 (DOPAC), $m/z$ 166 > 131 and 103 (Phe), $m/z$ 182 > 136 and 119 (Tyr) and $m/z$ 205 > 118 and 146 (Trp).

## Bone examination

The right femora was thawed, cleaned for soft connective tissue, and femoral length determined using a digital caliper. Subsequently, the femora were placed in a peripheral dual-emission x-ray absorptiometry (DEXA) scanner (Sabre XL; Norland Stratec) and scanned using a pixel resolution of $0.1 \times 0.1$ mm$^2$. The bone mineral content (BMC) and the areal bone mineral density (aBMD) of the whole bone were determined with the software provided by the MR system manufacturer. Quality assurance was performed by scans of the two solid-state phantoms provided with the scanner. The coefficient of variation (CV) of mice femoral aBMD was 2.6% (procedure repeated ten times on the same femur).

## Statistical analysis

All calculations were performed using the software SPSS 22 (SPSS; IBM) or Microsoft Excel 2010 for Windows. Paired- and unpaired-t-tests, Pearsons correlation test, one and two-way ANOVA performed two-sided at a significance level of $\alpha = 0.05$.

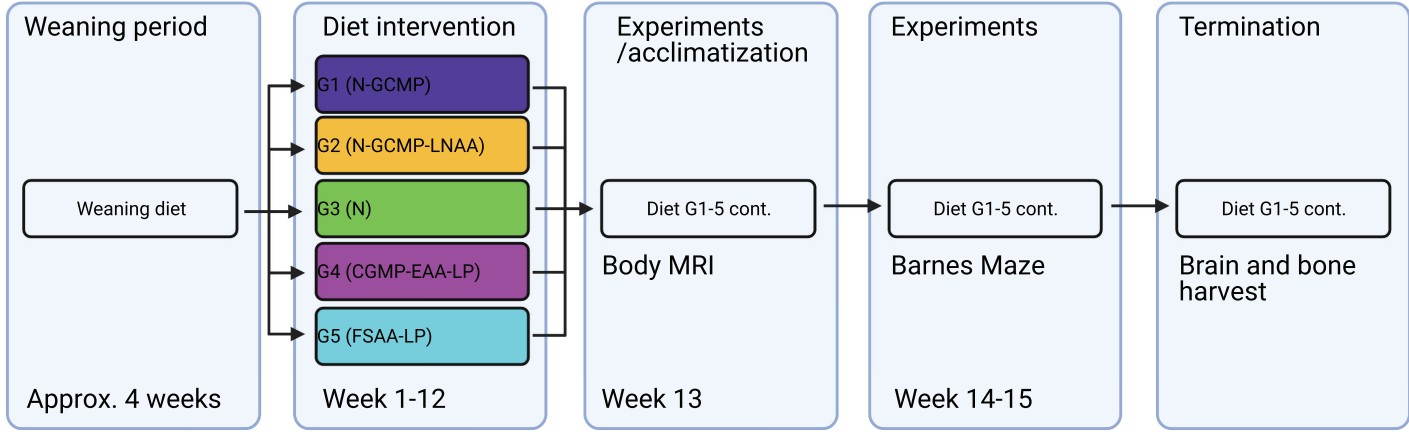

**Fig 1. Timeline of the study design (weeks refer to the week of diet treatment, which continuous from week 1 until week 15.**

### The experimental design

The experimental design was set to control for three main effects and their interactions: 1) effect of adding CGMP to a high Phe-diet (G1 (N-CGMP)); 2) adding CGMP to a high Phe-diet supplemented with Tyr, Leu and Trp (G2 (N-CGMP-LNAA)); 3) the effect of using CGMP as the main AA source in a LP diet (G4 (CGMP-EAA-LP)) compared to FSAA (G5 (FSAA-LP)). Mice fed at casein diet served as the control (G3(N)). After the weaning period for 28 days, female offspring were randomized to one of the five diets and kept on this diet through young adulthood until end of the experiment, which in total was a period of 15 weeks. The overall design of the study is shown with accompanied timeline in Fig 1.

## Results

Results for all 5 groups are presented in the same figures and tables, but statistically compared in sub-groups G1 (N-CGMP), G2 (N-CGMP-LNAA) and G3 (N) versus G3 (N), G4 (CGMP-EAA-LP) and G5 (FSAA-LP) when relevant, otherwise compared between all groups. G3 (N), untreated mice on casein diet, served as a control group.

### Food and water

Food intake were significantly higher for G2 (N-CGMP-LNAA) and G5 (FSAA-LP), when separately compared to G3 (N) (p = 0.01). Water: G2 (N-CGMP-LNAA) had an almost 2-fold increased intake of water compared to both G1 (N-CGMP) and G3 (N) (p < 0.01). G3 had the lowest intake of water in total but not significantly different from G4 (CGMP-EAA-LP) (p = 0.45). Intake of water in G5 (FSAA-LP) was significantly higher than in G4 (CGMP-EAA-LP) and G3 (N), when compared separately (p<0.01). Mean values (week1-11) of food and water intake pr. mouse/week (Mean ± SD) is shown in Fig 2A and 2B. There was a strong positive correlation between mean values of intake of food and water.

### Growth (weight start, weight end, weight gain) and body scan (fat, lean and weight)

As expected, due to randomization of the litter into the five groups, there was no significant difference between the start weight for the five groups. At the end of the study, G2 (N-CGMP-LNAA) had a significantly higher weight compared to G1 (N-CGMP) (p<0.05)

and G3 (N) (p<0.01). G4 (CGMP-EAA-LP) and G5 (FSAA-LP) were significantly higher for both weight end and weight gain compared to G3(N) (p ≤ 0.001). G4 (CGMP-EAA-LP) demonstrated the highest average growth compared to start weight (160.4%), followed by G5 (FSAA-LP) (146.5%) (Fig 2C). The same pattern was obtained for body composition of Lean (Fig 2D). G4 (CGMP-EAA-LP) and G5 (FSAA-LP) demonstrated the highest growth, indicating that CGMP and FSAA in combination with LP diet provide the best thrive. Our findings support the study by *Ney et al*, 2008 [12] showing a similar growth on CGMP compared to FSAA [12, 32].

## Bone density measurements/bone examination

Body scan supported the thrive results. Femur length (mm), (BMC) (g) and aBMD (g/cm$^2$) were significantly higher for G4 (CGMP-EAA-LP) and G5 (FSAA-LP) compared to G3(N). Furthermore, G2 (N-CGMP-LNAA) was significantly higher compared to G3 (N) for aBMD.

G4 (CGMP-EAA-LP) and G5 (FSAA-LP) revealed the highest value for BMC, aBMD and length of femora, which potentially indicate, that it is caused by the LP diet and supplement from either CGMP or FSAA, (p > 0.05 for all comparisons). All results are presented in Fig 2E–2G. Solverson et al documented similar increased bone strength in PKU mice after treatment with cGMP [32].

## Weekly Phe and Tyr plasma analyses (Dried Blood Spot (DBS)

The Phe concentration was lower in group G1 (N-CGMP) and G2 (N-CGMP-LNAA) compared to G3 (N) at all timepoints (week 1–11), but only at week 2, G3 (N) had significantly higher levels of plasma Phe compared to G1(N-CGMP). There was no significant difference in Phe concentrations between G3 (N) and G2 (N-CGMP-LNAA). By comparison of G1 (N-CGMP) with G2 (N-CGMP-LNAA) no significant difference in Phe concentration was observed, except at week 9, where G1 (N-CGMP) had significantly higher plasma Phe level compared to G2 (N-CGMP-LNAA).

At all timepoints, the Phe concentration in G4 (CGMP-EAA-LP) and G5 (FSAA-LP), were significantly lower compared to G3 (N). Significantly higher Phe concentration was observed in G5 (FSAA-LP) compared to G4 (CGMP-EAA-LP) at all time points, except for week 2, 4 and 11; Fig 3A

The concentrations of Tyr were similar for all groups, Fig 3B.

Due to lack of conversion of Phe to Tyr, affected individuals with PKU have an elevated Phe/Tyr ratio (typically >2.0 when untreated). The Phe/Tyr-ratio was significantly higher in G3 (N) compared to G4 (CGMP-EAA-LP), and G5 (FSAA-LP), Fig 3C.

## Total AA profile (week 6 and 12 of the diet intervention period)

It was not possible to take a blood test for determination of the total amino acid profile at week 1 of diet intervention since the mice at that time, were too small to tolerate blood sampling, but as all animals were treated equal before week 1, we assume that the start value was similar in all groups.

Total amino acid profile was obtained for all animals at week 6 and week 12 (Fig 4), including the LNAAs of special interest: Tyr, Trp, Leu (added extra to G2 (N-CGMP-LNAA), G4 (CGMP-EAA-LP) and G5 (FSAA-LP)), Ile, Thr, Val (CGMP contains high amounts of these AAs), Arg, His, Met, Lys (added extra in G4 (CGMP-EAA-LP) and G5 (FSAA-LP)) and Phe.

Comparison between groups at week 6 and at week 12, respectively, for the same AA revealed the following results: No significant differences between G1 (N-CGMP) and G2 (N-CGMP-LNAA) neither at week 6 or week 12 were observed, but compared to G3 (N), there

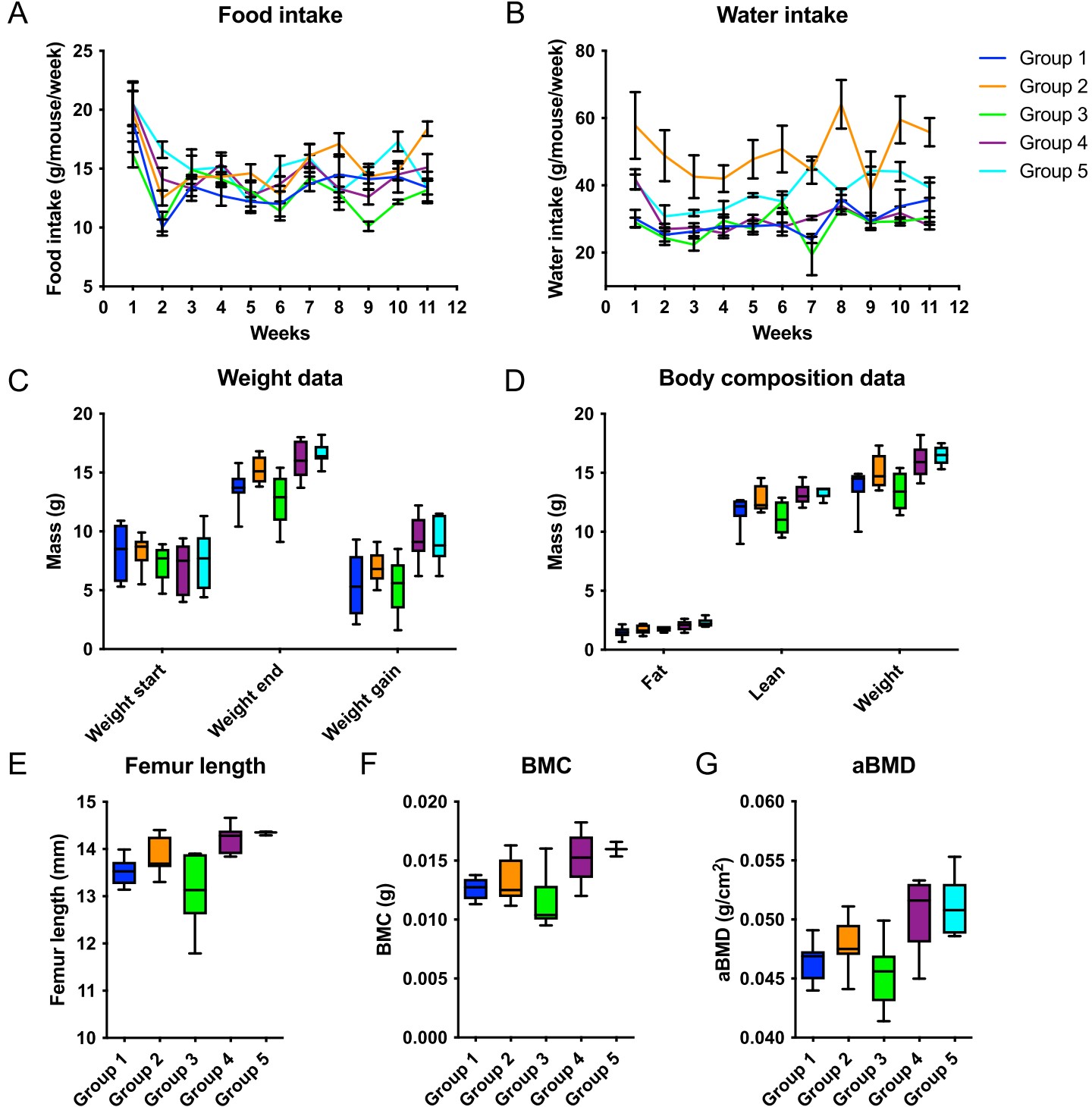

**Fig 2.** A: Average intake (g) of food pr. mouse/gr./w (w 1–11), Mean ± SD. G1 (13.6 +/- 3.1), G2 (15.4 +/- 2.5), G3 (13 (+/- 2.5), G2>G3 (p = 0.01), G4 (14.6 +/- 2.9), and G5 (15.4 +/- 2.8), G5>G3 (p = 0.01)). B: Average intake (g) of water pr. mouse/gr./w (w 1–11), Mean ± SD. G1 (29.5 (+/- 8.3); G2 (50.2 (+/- 20.3), G2>G1 (p< 0.001); G3 28.1 (+/- 6.9), G2>G3 (p< 0.001); G4 (30.3 (+/- 4.2), G5 (38.2 (+/- 5.7), G5>G3, G5>G4 (p≤0.001). C: Body Weight data (BW) (g) (weight start, weight end and weight gain) pr. mouse/gr./w (w1-12), Mean ± SD. Weight start: G1 (8.3 +/- 2.4), G2 (8.3 +/- 1.3), G3 (7.2 +/- 1.4), G4 (6.7 +/- 2.1) and G5 (7.4 +/- 2.5); Weight end: G1 (13.6 +/- 1.5), G2 (15.2 +/- 1.1), G3 (12.7 +/- 2.1), G4 (16.1 +/- 1.6) and G5 (16.6 +/- 0.9); Weight gain: G1 (5.4 +/- 2.5), G2 (6.9 +/- 1.3), G3 (5.4 +/- 2.3), G4 (9.3 +/- 1.9) and G5 (9.2 +/- 1.9). G2> G1 (p<0.05), G2> G3 (N) (p<0.01), G4 and G5 both > G3 (p ≤ 0.001). D: Body composition data, Body scan (g) (fat, lean body weight (LBW), weight total (WT)) pr. mouse/gr./w (w1-12) Mean ±SD. Fat: G1 (1.5 +/- 0.4), G2 (1.8 +/- 0.3), G3 (1.5 +/- 0.4), G4 (2.1 +/- 0.5), G5 (2.2 +/- 0.4), G4>G3 (p = 0.02), G5>G3 (p < 0.01)); LBW: G1 (12 +/- 1.3), G2 (13 +/- 1), G3 (10.8 +/- 1.5), G4 (13.5 +/- 0.9), G5 (13.7 +/- 0.7), G2>G3 (p < 0,01), G4>G3 (p < 0.001),

G5>G3 (p < 0.0001)); WT: G1 (14 +/- 1.7), G2 (15.3 +/- 1.2), G3 (12.9 +/- 1.9), G4 (16.3 +/- 1.4), G5 (16.7 +/- 1), G2>G3 (p < 0,01), G4>G3 (p < 0.001), G5>G3 (p < 0.0001). E-G: DEXA scanning of the right femora, presented as (E) length (mm), (F) The bone mineral content (BMC) (g), and (G) areal bone mineral density (aBMD) (g/cm2) of the right femora. Mean ± SD. E (length, mm): G1 (13.52 +/- 0.28), G2 (13.82 +/- 0.39), G3 (13.15 +/- 0.77), G4 (14.21 +/- 0.30) and G5 (14.34 +/- 0.04), G4>G3 (p < 0.01)), G5>G3 (p <0.05). FBMC (g): G1 (0.0126 +/- 0.0009), G2 (0.0133 +/- 0.0019), G3 (0.0116 +/- 0.0023), G4 (0.0152 +/- 0.0021) and G5 (0.0160 +/- 0.0009), G4>G3 (p = 0.01), G5>G3 (p < 0.05). G aBMD (g/cm2): G1 (0.0463 +/- 0.0016), G2 (0.0481 +/- 0.0021), G3 (0.0452 +/- 0.0026), G4 (0.0503 +/- 0.0029) and G5 (0.0511 +/- 0.0024), G2>G3 (p <0.05), G4>G3 (p = 0.001), G5>G3(p<0.001). G1-G5 represent G1(N-CGMP), G2 (N-CGMP-LNAA), G3 (N), G4 (CGMP-EAA-LP) and G5 (FSAA-LP).

were quite a few. Phe was significant lower in G1 and G2 (N-CGMP-LNAA) compared to G3 (N) at week 12, whereas Phe was significantly lower in G2 (N-CGMP-LNAA) (but not in G1 ((N-CGMP)) compared to G3 (N) at week 6. Furthermore, Thr was significantly higher in both G1 (N-CGMP) and G2 (N-CGMP-LNAA) at week 6 compared to G3 (N), whereas Trp was significantly higher in both G1 (N-CGMP) and G2 (N-CGMP-LNAA) compared to G3 (N) at week 12. In addition, Thr and Leu was also significantly higher in G2 (N-CGMP-LNAA), but not in G1 (N-CGMP) compared to G3 (N) at week 12.

It is possible that the increased amount of Lys in G2 (N-CGMP-LNAA), compared to G3 (N), although only significant level was obtained at week 12, might contribute to the significantly lower Phe level in G2 (N-CGMP-LNAA) already at week 6 compared to G3 (N).

By comparison of G3 (N), G4 (CGMP-EAA-LP) and G5 (FSAA-LP), we found that Phe was significantly higher in G3 (N) compared to G4 (CGMP-EAA-LP) and G5 (FSAA-LP) both at week 6 and at week 12, and Phe was significantly higher in G5 (FSAA-LP) compared to G4 (CGMP-EAA-LP) at week 12. The AA Ile and Leu was significantly higher in G4 (CGMP-EAA-LP) compared to G5 (FSAA-LP) both at week 6 and at week 12. These might contribute to the higher plasma concentration in G5 (FSAA-LP). P-values for comparison of subgroups are presented in Table 3

Also, the change from week 6 to week 12 were investigated for the different subgroups. By comparing changes for the individual AA's between week 6 and 12, Trp increased significantly in G1(N-CGMP) and G2(N-CGMP-LNAA) from week 6 to week 12, and Leu increased significantly in G2(N-CGMP-LNAA).

Most significant changes were found for G4 (CGMP-EAA-LP) and G5 (FSAA-LP). Ile and Val increased significantly in both groups from week 6 to week 12 which could potentially be due to the high content of these AA in CGMP in G4 and as FSAA in G5, but also the majority

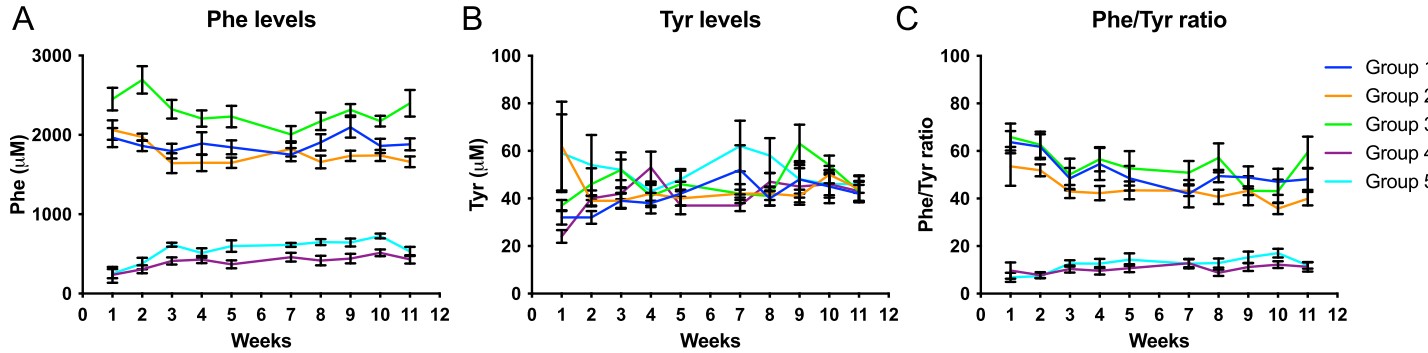

**Fig 3. Average plasma Phe, plasma Tyr- and plasma Phe/Tyr-ratio (µmol/l), mean ± SD.** G1-5/w1-11. A: Phe levels. G3>G4 and G5 at all time points (p < 0.01), G3>G1 at w2 (p <0.05), G5>G4 at all time points except w2, 4 and 11 (p <0.05). B: Tyr levels. G1 (0.0126 ± 0.0009), G2 (0.0133 ± 0.0019), G3 (0.0116 ± 0.0023), G4 (0.0152 ± 0.0021) and G5 (0.0160 ± 0.0009). W1: G3>G4, w7: G5>G4, w9: G3>G2 (p < 0.01). C: Phe/Tyr levels. G1 (0.0463 ± 0.0016), G2 (0.0481 ± 0.0021), G3 (0.0452 ±0.0026), G4 (0.0503 ± 0.0029) and G5 (0.0511 ± 0.0024),(G2>G3 (p <0.05), G4>G3 (p = 0.001), G5>G3 (p<0.001). Group 1–5 represent: G1(N-CGMP), G2 (N-CGMP-LNAA), G3 (N), G4 (CGMP-EAA-LP) and G5 (FSAA-LP).

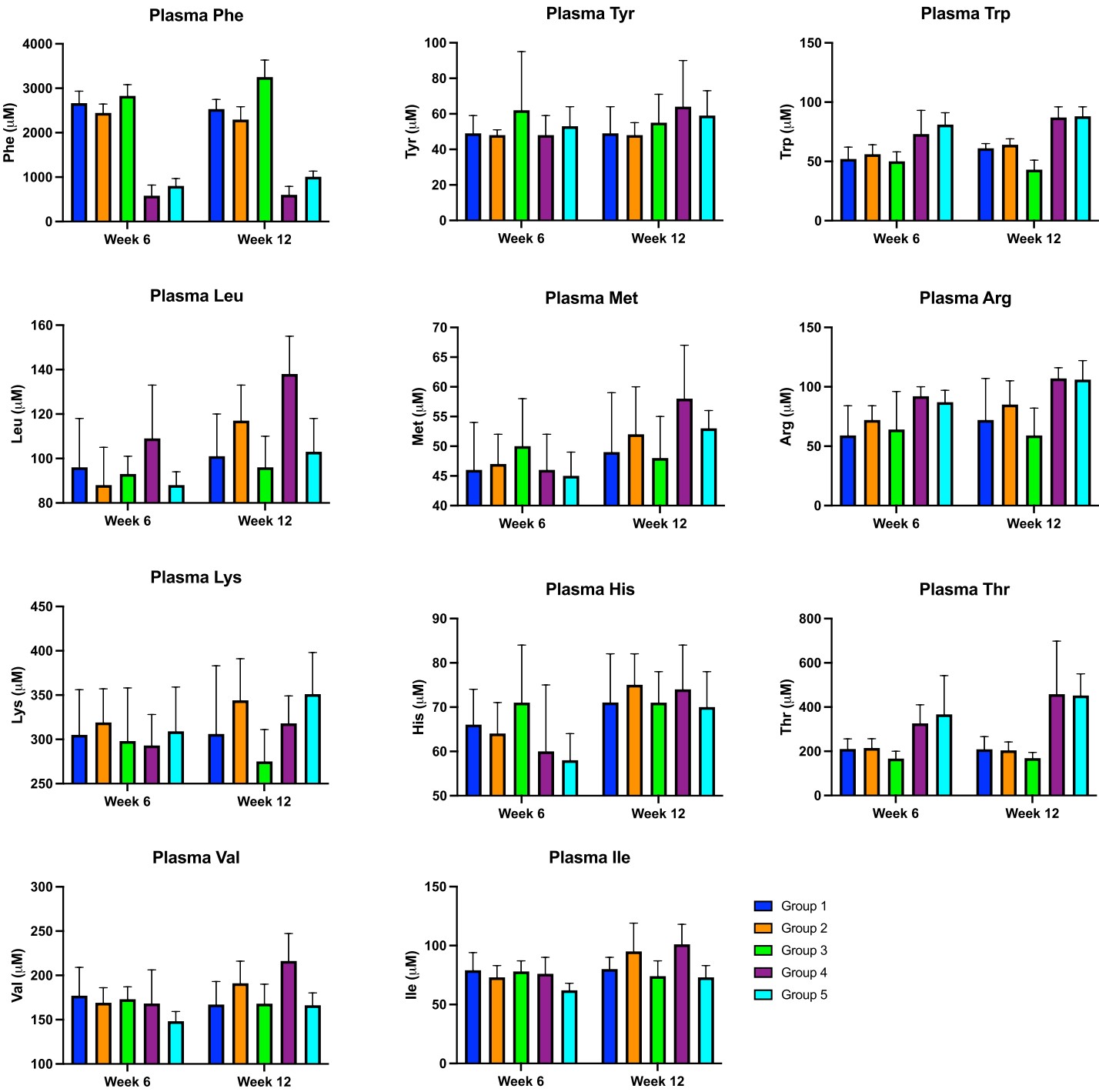

**Fig 4. Average plasma concentrations of AA at w6 and w12.** Group 1–5 represent: G1(N-CGMP), G2 (N-CGMP-LNAA), G3 (N), G4 (CGMP-EAA-LP) and G5 (FSAA-LP).

of the extra added AAs (Leu, Met, His and Arg) in G4 (CGMP-EAA-LP) and G5 (FSAA-LP) increased significantly ($p < 0.05$).

As expected, Phe increased significantly for G3 (N), but surprisingly also for G5 (FSAA-LP), which theoretically could be related to the slightly higher intake of Phe-containing water; 3.11

**Table 3. P-values for comparison of the different groups G1-G5, representing G1(N-CGMP), G2 (N-CGMP-LNAA), G3 (N), G4 (CGMP-EAA-LP) and G5 (FSAA-LP).**

| Week 6 | G1 & G2 | G1 & G3 | | G2 & G3 | | G4 & G5 | | G4 & G3 | | G5 & G3 | |
|---|---|---|---|---|---|---|---|---|---|---|---|
| Lys | ns | ns | | ns | | ns | | ns | | ns | |
| Arg | ns | ns | | ns | | ns | | p < 0.05 | G4>G3 | ns | |
| His | ns | ns | | ns | | ns | | ns | | p < 0.05 | G5<G3 |
| Trp | ns | ns | | ns | | ns | | p < 0.01 | G4>G3 | p < 0.01 | G5>G3 |
| Thr | ns | p < 0.05 | G1>G3 | p < 0.05 | G2>G3 | ns | | p < 0.01 | G4>G3 | p < 0.01 | G5>G3 |
| Val | ns | ns | | ns | | ns | | ns | | p < 0.01 | G5<G3 |
| Met | ns | ns | | ns | | ns | | ns | | ns | |
| Ile | ns | ns | | ns | | p < 0.05 | G4>G5 | ns | | p < 0.01 | |
| Leu | ns | ns | | ns | | p < 0.05 | G4>G5 | ns | | ns | |
| Tyr | ns | ns | | ns | | ns | | ns | | ns | |
| Phe | ns | ns | | p < 0.01 | G3>G2 | p = 0.05 | | p < 0.01 | G4<G3 | p < 0.01 | G5<G3 |
| Week 12 | G1 & G2 | G1 & G3 | | G2 & G3 | | G4 & 5 | | G4 & G3 | | G5 & G3 | |
| Lys | ns | ns | | p < 0.01 | G2>G3 | ns | | p < 0.05 | | p < 0.01 | G5>G3 |
| Arg | ns | ns | | p < 0.05 | G2>G3 | ns | | p < 0.01 | G4>G3 | p < 0.01 | G5>G3 |
| His | ns | ns | | ns | | ns | | ns | | ns | |
| Trp | ns | p < 0.01 | G1>G3 | p < 0.01 | G2>G3 | ns | | p < 0.01 | G4>G3 | p < 0.01 | G5>G3 |
| Thr | ns | ns | | p < 0.05 | G2>G3 | ns | | p < 0.01 | G4>G3 | p < 0.01 | G5>G3 |
| Val | ns | ns | | ns | | p < 0.01 | G4>G5 | p < 0.01 | G4>G3 | ns | |
| Met | ns | ns | | ns | | ns | | p < 0.05 | G4>G3 | ns | |
| Ile | ns | ns | | ns | | p < 0.01 | G4>G5 | p < 0.01 | G4>G3 | ns | |
| Leu | ns | ns | | p < 0.05 | G2>G3 | p < 0.01 | G4>G5 | p < 0.01 | G4>G3 | ns | |
| Tyr | ns | ns | | ns | | ns | | ns | | ns | |
| Phe | ns | p < 0.01 | G1<G3 | p < 0.01 | G2<G3 | p < 0.01 | G4<G5 | p < 0.01 | G4<G3 | p < 0.01 | G5<G3 |

Significant results are highlighted.

mg/day for G5 (FSAA-LP) compared to 2.47 mg/day for G4 (CGMP-EAA-LP). However, this did not prevent the other LNAA from increasing also. Mean intake of food varied from 13.0 (G3) to 15.4 (G5) g/week. Concentrations of selected AA in plasma presented as mean with SD for week 6 (sample 1) and week 12 (sample 2), difference between sample 1 and 2 and %-change and p-values for comparison of week 6 and week 12 are presented in Table 4.

**Brain examination.** The following brain parts were examined: Cerebellum, brain stem, hypothalamus, parietal cortex, anterior piriform, cortex and olfactory bulb. We tested for the following AA and their metabolites: Phe, Tyr, Trp, 5-HT, 5-HIAA, DA and DOPAC. The amino acids, Trp, Tyr and Phe utilize the same LAT1 transporter over the BBB.

Trp is metabolized to 5-HT in the brain and finally further metabolized mainly to 5-HIAA. Tyr is precursor of DA which is further converted to DOPAC. These metabolites are crucial in order to evaluate the LNAA balance in the brain.

The content of DA, 5-HT, 5-HIAA, DOPAC, Phe, Tyr and Trp (Mean ± SD) for total brain, by ading values for the individual brain parts together is presented in Fig 5.

Overall, G3 (N) had the highest Phe levels and G4 (CGMP-EAA-LP) had the lowest in the entire brain followed by G5 (FSAA-LP). The concentration of Phe in G3 (N) was significantly increased compared to all the other groups.

In contrast, Tyr in total brain levels only showed minor variations between the groups despite the different content in the five chow (0.67–1.44 g AA/100 g).

**Table 4. Plasma content of selected AA in plasma measured at week 6 and 12 presented as mean and SD, difference from start to end and % change.**

| ID (group) | Week 6 (μM/l) | Week 12 (μM/l) | difference | % change | p-value |
|---|---|---|---|---|---|
| | | Phe | | | |
| 1 | 2665 +/- 270 | 2533 +/- 214 | -132 | -5 | ns |
| 2 | 2444 +/- 200 | 2296 +/- 289 | -149 | -6 | ns |
| 3 | 2828 +/- 253 | 3253 +/- 381 | 425 | 15 | p = 0.01 |
| 4 | 580 +/- 243 | 601 +/- 189 | 21 | 4 | ns |
| 5 | 800 +/- 167 | 1009 +/- 126 | 209 | 26 | p = 0.01 |
| | | Tyr | | | |
| 1 | 49 +/- 10 | 49 +/- 15 | 0 | 0 | ns |
| 2 | 48 +/- 3 | 48 +/- 7 | 0 | 0 | ns |
| 3 | 62 +/- 33 | 55 +/- 16 | -7 | -11 | ns |
| 4 | 48 +/- 11 | 64 +/- 26 | 17 | 35 | ns |
| 5 | 53 +/- 11 | 59 +/- 14 | 6 | 11 | ns |
| | | Trp | | | |
| 1 | 52 +/- 10 | 61 +/- 4 | 9 | 17 | p<0.05 |
| 2 | 56 +/- 8 | 64 +/- 5 | 8 | 15 | p<0.05 |
| 3 | 50 +/- 8 | 43 +/- 8 | -7 | -13 | ns |
| 4 | 73 +/- 20 | 87 +/- 9 | 14 | 19 | ns |
| 5 | 81 +/- 10 | 88 +/- 8 | 7 | 9 | ns |
| | | Leu | | | |
| 1 | 96 +/- 22 | 101 +/- 19 | 5 | 5 | ns |
| 2 | 88 +/- 17 | 117 +/- 16 | 29 | 33 | p<0.01 |
| 3 | 93 +/- 8 | 96 +/- 14 | 3 | 3 | ns |
| 4 | 109 +/- 24 | 138 +/- 17 | 29 | 26 | p = 0.01 |
| 5 | 88 +/- 6 | 103 +/- 15 | 15 | 17 | p<0.05 |
| | | Met | | | |
| 1 | 46 +/- 8 | 49 +/- 10 | 3 | 6 | ns |
| 2 | 47 +/- 5 | 52 +/- 8 | 5 | 12 | ns |
| 3 | 50 +/- 8 | 48 +/- 7 | -2 | -4 | ns |
| 4 | 46 +/- 6 | 58 +/- 9 | 12 | 27 | p<0.01 |
| 5 | 45 +/- 4 | 53 +/- 3 | 8 | 17 | p<0.01 |
| | | Arg | | | |
| 1 | 59 +/- 25 | 72 +/- 35 | 14 | 23 | ns |
| 2 | 72 +/- 12 | 85 +/- 20 | 13 | 18 | ns |
| 3 | 64 +/- 32 | 59 +/- 23 | -6 | -9 | ns |
| 4 | 92 +/- 8 | 107 +/- 9 | 15 | 16 | p<0.01 |
| 5 | 87 +/- 10 | 106 +/- 16 | 18 | 21 | p = 0.01 |
| | | Lys | | | |
| 1 | 305 +/- 51 | 306 +/- 77 | 1 | 0 | ns |
| 2 | 319 +/- 38 | 344 +/- 47 | 25 | 8 | ns |
| 3 | 298 +/- 60 | 275 +/- 36 | -23 | -8 | ns |
| 4 | 293 +/- 35 | 318 +/- 31 | 24 | 8 | ns |
| 5 | 309 +/- 50 | 351 +/- 47 | 41 | 13 | ns |
| | | His | | | |
| 1 | 66 +/- 8 | 71 +/- 11 | 4 | 7 | ns |
| 2 | 64 +/- 7 | 75 +/- 7 | 11 | 18 | p<0.01 |
| 3 | 71 +/- 13 | 71 +/- 7 | 0 | 0 | ns |
| 4 | 60 +/- 15 | 74 +/- 10 | 15 | 25 | p<0.05 |

*(Continued)*

**Table 4.** (Continued)

| ID (group) | Week 6 (µM/l) | Week 12 (µM/l) | difference | % change | p-value |
|---|---|---|---|---|---|
| 5 | 58 +/- 6 | 70 +/- 8 | 12 | 21 | p<0.01 |
| | | Thr | | | |
| 1 | 210 +/- 46 | 209 +/- 57 | -1 | 0 | ns |
| 2 | 215 +/- 42 | 204 +/- 38 | -10 | -5 | ns |
| 3 | 167 +/- 33 | 169 +/- 26 | 2 | 1 | ns |
| 4 | 326 +/- 84 | 458 +/- 240 | 132 | 40 | ns |
| 5 | 367 +/- 175 | 452 +/- 98 | 84 | 23 | ns |
| | | Val | | | |
| 1 | 177 +/- 32 | 167 +/- 26 | -10 | -6 | ns |
| 2 | 169 +/- 17 | 191 +/- 25 | 22 | 13 | ns |
| 3 | 173 +/- 14 | 168 +/- 22 | -5 | -3 | ns |
| 4 | 168 +/- 38 | 216 +/- 31 | 48 | 29 | p = 0.01 |
| 5 | 148 +/- 11 | 166 +/- 14 | 18 | 12 | p = 0.01 |
| | | Ile | | | |
| 1 | 79 +/- 15 | 80 +/- 10 | 0 | 0 | ns |
| 2 | 73 +/- 10 | 95 +/- 24 | 22 | 30 | p<0.05 |
| 3 | 78 +/- 9 | 74 +/- 13 | -4 | -5 | ns |
| 4 | 76 +/- 14 | 101 +/- 17 | 25 | 33 | p<0.01 |
| 5 | 62 +/- 6 | 73 +/- 10 | 12 | 19 | p<0.05 |

Group 1–5 represent G1(N-CGMP), G2 (N-CGMP-LNAA), G3 (N), G4 (CGMP-EAA-LP) and G5 (FSAA-LP).

Trp levels were significantly higher in G1 (N-CGMP) compared to G2 (N-CGMP-LNAA), and significantly higher in G5 (FSAA-LP) compared to G3 (N) and G4 (CGMP-EAA-LP). DOPAC only reached significant increased level in G5 (FSAA-LP) compared to G3 (N) ($p < 0.05$). DA did not reach significant level at all. 5-HT and 5-HIAA both demonstrated a 2-3-fold higher value in G4 and G5 (FSAA-LP) compared to G1(N-CGMP), G2 (N-CGMP-LNAA), and G3 (N).

Investigation of the individual brain parts revealed a significant effect compared to control G3(N) for all groups on most brain parts for Phe. G1 (N-CGMP) and G2(N-CGMP-LNAA) had almost similar levels in all brain parts, and no significant differences were found (Fig 6).

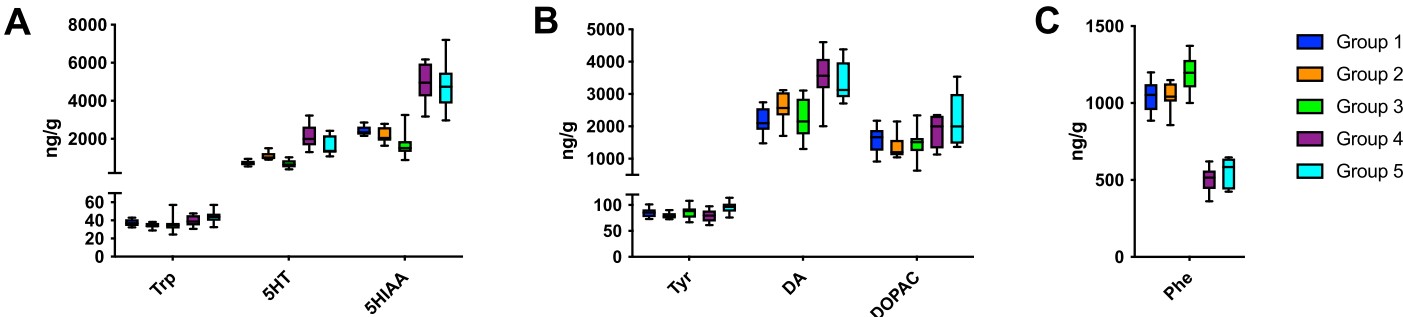

**Fig 5. The total content of DA, 5-HT, 5-HIAA, DOPAC, Phe, Tyr and Trp from the following brain parts cerebellum, brain stem, hypothalamus, parietal cortex, anterior piriform, cortex and olfactory bulb (presented as total brain by ading values for the individual brain parts together).** Trp is metabolized to 5-HT in the brain and finally further metabolized mainly to 5-HIAA. Tyr is precursor of DA which is further converted to DOPAC. Group 1–5 represent: G1(N-CGMP), G2 (N-CGMP-LNAA), G3 (N), G4 (CGMP-EAA-LP) and G5 (FSAA-LP).

The concentration of Tyr and Trp (34) in the brain theoretically determine the brain concentration of DA and 5-HT respectively. Both Trp and Tyr levels were significantly increased in the brain stem (BS) in G1(N-CGMP) compared to G2(N-CGMP-LNAA). In spite of this, 5HT was significantly increased in several brain parts (anterior piriform (AP), brain stem (BS), hypothalamus (HT) in G2 (N-CGMP-LNAA) compared to G1 (N-CGMP). Also, significantly higher levels of 5-HT in G2 (N-CGMP-LNAA) compared to G3(N) were observed in most brain parts. The concentration of 5-HIAA was increased significantly in several brain parts in both G1(N-CGMP) and G2 (N-CGMP-LNAA) compared to G3(N). P-values are presented in Table 5. Thus, no clear correlation between the Trp, 5HT and 5-HIAA concentration exist in G1(N-CGMP) and G2 (N-CGMP-LNAA). In contrast, the higher concentration of Trp in G4 (CGMP-EAA-LP) and G5 (FSAA-LP) compared to G3 (N) also leads to higher 5-HT concentration in G4 (CGMP-EAA-LP) and G5 (FSAA-LP). Also, the concentration of 5HIAA was higher in the brain of G4 (CGMP-EAA-LP) and G5 (FSAA-LP), compared to G3 (N).

Although the concentration of Tyr was significantly increased in several brain parts in G5 (FSAA-LP) compared to G4 (CGMP-EAA-LP) this was not associated with an increase in neither the concentration of DA nor DOPAC (Fig 6).

Comparison of the total brain concentration and plasma concentration of Phe indicated a strong correlation between the concentration in the blood and the brain (Fig 7A). Also, the concentration of Trp in plasma and brain indicate correlation (Fig 7C). A high concentration of Trp in the blood were associated with a high concentration in the brain. For Tyr there was no clear correlation (Fig 7C).

The plasma Phe values were in line with the concentration-levels found in the total brain, indicating, that the concentration in plasma is a major factor determining the concentration of Phe in the brain while the relation between plasma- and brain-concentration of Tyr and Trp, respectively, is determined by other factors, as previously documented by Berry, Harding and Pacussi [33–35].

No clear correlation could be observed when investigation of the brain concentration of Trp with the brain concentration of 5-HT (Fig 7D). 5-HIAA is the main metabolite of 5-HT [36] and we did find a strong correlation (0.87) (Fig 7E). Despite this, the correlation between the concentration of 5-HT and 5-HIAA in G4 (CGMP-EAA-LP) and G5 (FSAA-LP) was not obvious (Fig 7E). Investigation of the brain concentration of Tyr and DA and of DA and DOPAC revealed no correlation, as expected (Fig 7F and 7G).

The lack of clear correlations between the concentration of Trp and Tyr and its metabolites, are in agreement with observations published by Harding et al., argued that the most likely cause of brain DA and 5-HT deficiency in PKU is Phe-mediated inhibition of brain Tyr hydroxylase (TH) and Trp hydroxylase (TPH) activities rather than decreased substrate availability [35]. The lack of any significant difference according to the concentration of DA, is consistent with the findings of Puglisi-Allegra et al., demonstrating that DA is the least affected brain amine in PKU mice [37].

**Behavioral test (Barnes maze).** Metrics measured for the behavioral studies were: 1) Duration of each trial (Fig 8A) 2) the mean speed of the animal (Fig 8B) and 3) the distance travelled (Fig 8C). Unlike test duration, the mean speed of the animals did not seem to alter during testing. See legend, Fig 8 for significant differences.

## Exclusion of animals with malocclusion due to incisor overgrowth

A high incidence of malocclusion due to overgrowth of the incisors was found in this type of PKU mice. The regular contact due to body weight measurement made it possible to diagnose the presence of these lesions, and a programme of regular tooth trimming using blunt-tipped

**Table 5. Difference in concentrations of selected AA and metabolites in different brain parts.**

| | Anterior piriform | | | Brain stem | | | Bulbus olfactorius | | | Cerebellum | | | Hypothalamus | | | Parietal cortex | | |
|---|---|---|---|---|---|---|---|---|---|---|---|---|---|---|---|---|---|---|
| | compound | p value | | compound | p value | | compound | p value | | compound | p value | | compound | p value | | compound | p value | |
| 1,2 | Phe | 0.7879 | | Phe | 0.7998 | | Phe | 0.8945 | | Phe | 0.8953 | | Phe | 0.2752 | | Phe | 0.5250 | |
| 1,2 | Tyr | 0.7564 | | Tyr | 0.0085 | 1 > 2 | Tyr | 0.6638 | | Tyr | 0.0970 | | Tyr | 0.5201 | | Tyr | 0.1666 | |
| 1,2 | Trp | 0.6513 | | Trp | 0.0116 | 1 > 2 | Trp | 0.5298 | | Trp | 0.2778 | | Trp | 0.3170 | | Trp | 0.2335 | |
| 1,2 | DA | 0.1887 | | DA | 0.2201 | | DA | 0.2061 | | DA | 0.6869 | | DA | 0.0563 | | DA | 0.5618 | |
| 1,2 | .5-HT | 0.0012 | 2 > 1 | .5-HT | 0.0044 | 2 > 1 | .5-HT | 0.0297 | | .5-HT | 0.7493 | | .5-HT | 0.0345 | 2 > 1 | .5-HT | 0.6193 | |
| 1,2 | 5-HIAA | 0.8853 | | 5-HIAA | 0.7697 | | 5-HIAA | 0.0641 | | 5-HIAA | 0.0311 | 2 > 1 | 5-HIAA | 0.3304 | | 5-HIAA | 0.2547 | |
| 1,2 | DOPAC | 0.8451 | | DOPAC | 0.8805 | | DOPAC | 0.1396 | | DOPAC | 0.2795 | | DOPAC | 0.9644 | | DOPAC | 0.1718 | |
| 1,3 | Phe | 0.0133 | 3 > 1 | Phe | 0.0004 | 3 > 1 | Phe | 0.0186 | 3 > 1 | Phe | 0.0025 | 3 > 1 | Phe | 0.0845 | | Phe | 0.0043 | 3 > 1 |
| 1,3 | Tyr | 0.8724 | | Tyr | 0.1502 | | Tyr | 0.3975 | | Tyr | 0.8367 | | Tyr | 0.9032 | | Tyr | 0.4341 | |
| 1,3 | Trp | 0.7648 | | Trp | 0.4980 | | Trp | 0.7689 | | Trp | 0.8210 | | Trp | 0.7913 | | Trp | 0.5858 | |
| 1,3 | DA | 0.0711 | | DA | 0.2912 | | DA | 0.2695 | | DA | 0.9945 | | DA | 0.7134 | | DA | 0.3549 | |
| 1,3 | .5-HT | 0.8222 | | .5-HT | 0.2320 | | .5-HT | 0.8345 | | .5-HT | 0.8833 | | .5-HT | 0.5810 | | .5-HT | 0.0290 | 1 > 3 |
| 1,3 | 5-HIAA | 0.0695 | | 5-HIAA | 0.9944 | | 5-HIAA | 0.2008 | | 5-HIAA | 0.0115 | 1 > 3 | 5-HIAA | 0.0346 | | 5-HIAA | 0.0114 | 1 > 3 |
| 1,3 | DOPAC | 0.7140 | | DOPAC | 0.9410 | | DOPAC | 0.3591 | | DOPAC | 0.6047 | | DOPAC | 0.9589 | | DOPAC | 0.4987 | |
| 2,3 | Phe | 0.0408 | 3 > 2 | Phe | 0.0006 | 3 > 2 | Phe | 0.0560 | | Phe | 0.0003 | 3 > 2 | Phe | 0.0056 | 3 > 2 | Phe | 0.0041 | 3 > 2 |
| 2,3 | Tyr | 0.9308 | | Tyr | 0.0037 | 3 > 2 | Tyr | 0.6715 | | Tyr | 0.0990 | | Tyr | 0.5060 | | Tyr | 0.5123 | |
| 2,3 | Trp | 0.9542 | | Trp | 0.2269 | | Trp | 0.4101 | | Trp | 0.4390 | | Trp | 0.6717 | | Trp | 0.8999 | |
| 2,3 | DA | 0.8181 | | DA | 0.6245 | | DA | 0.8394 | | DA | 0.6265 | | DA | 0.0248 | 2 > 3 | DA | 0.8466 | |
| 2,3 | .5-HT | 0.0002 | 2 > 3 | .5-HT | 0.0814 | | .5-HT | 0.0371 | 2 > 3 | .5-HT | 0.5343 | | .5-HT | 0.0024 | 2 > 3 | .5-HT | 0.0012 | 2 > 3 |
| 2,3 | 5-HIAA | 0.0018 | 2 > 3 | 5-HIAA | 0.8356 | | 5-HIAA | 0.0153 | 2 > 3 | 5-HIAA | 0.2328 | | 5-HIAA | 0.0356 | 2 > 3 | 5-HIAA | 0.0018 | 2 > 3 |
| 2,3 | DOPAC | 0.6253 | | DOPAC | 0.8220 | | DOPAC | 0.4868 | | DOPAC | 0.4190 | | DOPAC | 0.9860 | | DOPAC | 0.3472 | |
| 3,4 | Phe | 0.0000 | 3 > 4 | Phe | 0.0000 | 3 > 4 | Phe | 0.0000 | 3 > 4 | Phe | 0.0000 | 3 > 4 | Phe | 0.0000 | 3 > 4 | Phe | 0.0000 | 3 > 4 |
| 3,4 | Tyr | 0.8089 | | Tyr | 0.0025 | 4 > 3 | Tyr | 0.8022 | | Tyr | 0.1360 | | Tyr | 0.3538 | | Tyr | 0.9469 | |
| 3,4 | Trp | 0.1889 | | Trp | 0.8038 | | Trp | 0.3170 | | Trp | 0.9670 | | Trp | 0.1782 | | Trp | 0.1453 | |
| 3,4 | DA | 0.1572 | | DA | 0.6240 | | DA | 0.8389 | | DA | 0.0301 | 4 > 3 | DA | 0.1563 | | DA | 0.0046 | 4 > 3 |
| 3,4 | .5-HT | 0.0000 | 4 > 3 | .5-HT | 0.0024 | 4 > 3 | .5-HT | 0.0003 | 4 > 3 | .5-HT | 0.0021 | 4 > 3 | .5-HT | 0.0001 | 4 > 3 | .5-HT | 0.0000 | 4 > 3 |
| 3,4 | 5-HIAA | 0.0000 | 4 > 3 | 5-HIAA | 0.0001 | 4 > 3 | 5-HIAA | 0.0001 | 4 > 3 | 5-HIAA | 0.0014 | 4 > 3 | 5-HIAA | 0.0000 | 4 > 3 | 5-HIAA | 0.0000 | 4 > 3 |
| 3,4 | DOPAC | 0.4708 | | DOPAC | 0.8229 | | DOPAC | 0.8374 | | DOPAC | 0.2559 | | DOPAC | 0.7765 | | DOPAC | 0.0648 | |
| 3,5 | Phe | 0.0000 | 3 > 5 | Phe | 0.0000 | 3 > 5 | Phe | 0.0000 | 3 > 5 | Phe | 0.0000 | 3 > 5 | Phe | 0.0000 | 3 > 5 | Phe | 0.0000 | 3 > 5 |
| 3,5 | Tyr | 0.0061 | 5 > 3 | Tyr | 0.2414 | | Tyr | 0.3133 | | Tyr | 0.4329 | | Tyr | 0.2761 | | Tyr | 0.0132 | 5 > 3 |
| 3,5 | Trp | 0.0127 | 5 > 3 | Trp | 0.0978 | | Trp | 0.2009 | | Trp | 0.4123 | | Trp | 0.0421 | | Trp | 0.0331 | 5 > 3 |
| 3,5 | DA | 0.9627 | | DA | 0.9576 | | DA | 0.5314 | | DA | 0.1247 | | DA | 0.0110 | 5 > 3 | DA | 0.0011 | 5 > 3 |
| 3,5 | .5-HT | 0.0001 | 5 > 3 | .5-HT | 0.1094 | | .5-HT | 0.0022 | 5 > 3 | .5-HT | 0.0010 | 5 > 3 | .5-HT | 0.0003 | 5 > 3 | .5-HT | 0.0000 | 5 > 3 |
| 3,5 | 5-HIAA | 0.0000 | 5 > 3 | 5-HIAA | 0.0019 | 5 > 3 | 5-HIAA | 0.0000 | 5 > 3 | 5-HIAA | 0.0006 | 5 > 3 | 5-HIAA | 0.0000 | 5 > 3 | 5-HIAA | 0.0000 | 5 > 3 |
| 3,5 | DOPAC | 0.1917 | | DOPAC | 0.1836 | | DOPAC | 0.1790 | | DOPAC | 0.0044 | 5 > 3 | DOPAC | 0.1430 | | DOPAC | 0.0501 | 5 > 3 |
| 4,5 | Phe | 0.1217 | | Phe | 0.2863 | | Phe | 0.8234 | | Phe | 0.3462 | | Phe | 0.6233 | | Phe | 0.2721 | |
| 4,5 | Tyr | 0.0019 | 5 > 4 | Tyr | 0.0090 | 5 > 4 | Tyr | 0.2326 | | Tyr | 0.0146 | 5 > 4 | Tyr | 0.0341 | 5 > 4 | Tyr | 0.0135 | 5 > 4 |

(*Continued*)

**Table 5.** (Continued)

| | Anterior piriform | | | Brain stem | | | Bulbus olfactorius | | | Cerebellum | | | Hypothalamus | | | Parietal cortex | | |
|---|---|---|---|---|---|---|---|---|---|---|---|---|---|---|---|---|---|---|
| | compound | p value | | compound | p value | | compound | p value | | compound | p value | | compound | p value | | compound | p value |
| 4,5 | Trp | 0.0439 | 5 > 4 | Trp | 0.0320 | 5 > 4 | Trp | 0.6502 | | Trp | 0.2831 | | Trp | 0.2502 | | Trp | 0.2713 |
| 4,5 | DA | 0.1525 | | DA | 0.6086 | | DA | 0.6899 | | DA | 0.5418 | | DA | 0.6666 | | DA | 0.8516 |
| 4,5 | .5-HT | 0.1272 | | .5-HT | 0.1041 | | .5-HT | 0.0489 | 4 > 5 | .5-HT | 0.3136 | | 5-HT | 0.3208 | | .5-HT | 0.0870 |
| 4,5 | 5-HIAA | 0.1394 | | 5-HIAA | 0.1556 | | 5-HIAA | 0.0277 | 4 > 5 | 5-HIAA | 0.2488 | | 5-HIAA | 0.5349 | | 5-HIAA | 0.6324 |
| 4,5 | DOPAC | 0.5633 | | DOPAC | 0.1696 | | DOPAC | 0.2185 | | DOPAC | 0.0378 | 5 > 4 | DOPAC | 0.2968 | | DOPAC | 0.5607 |

Significant results for comparisons between groups for Phe (μg/g), Tyr (μg/g), Trp (μg/g), Trp (μg/g), DA (ng/g), 5-HIAA(ng/g), DOPAC(ng/g) and 5-HT'(ng/g) in all the individual brain parts (anterior piriform (AP), brain stem (BS), olfactory bulb (BO), Cerebellum (CE), hypothalamus (HT) & parietal cortex (PC). The numbers 1–5 represent G1(N-CGMP), G2 (N-CGMP-LNAA), G3 (N), G4 (CGMP-EAA-LP) and G5 (FSAA-LP).

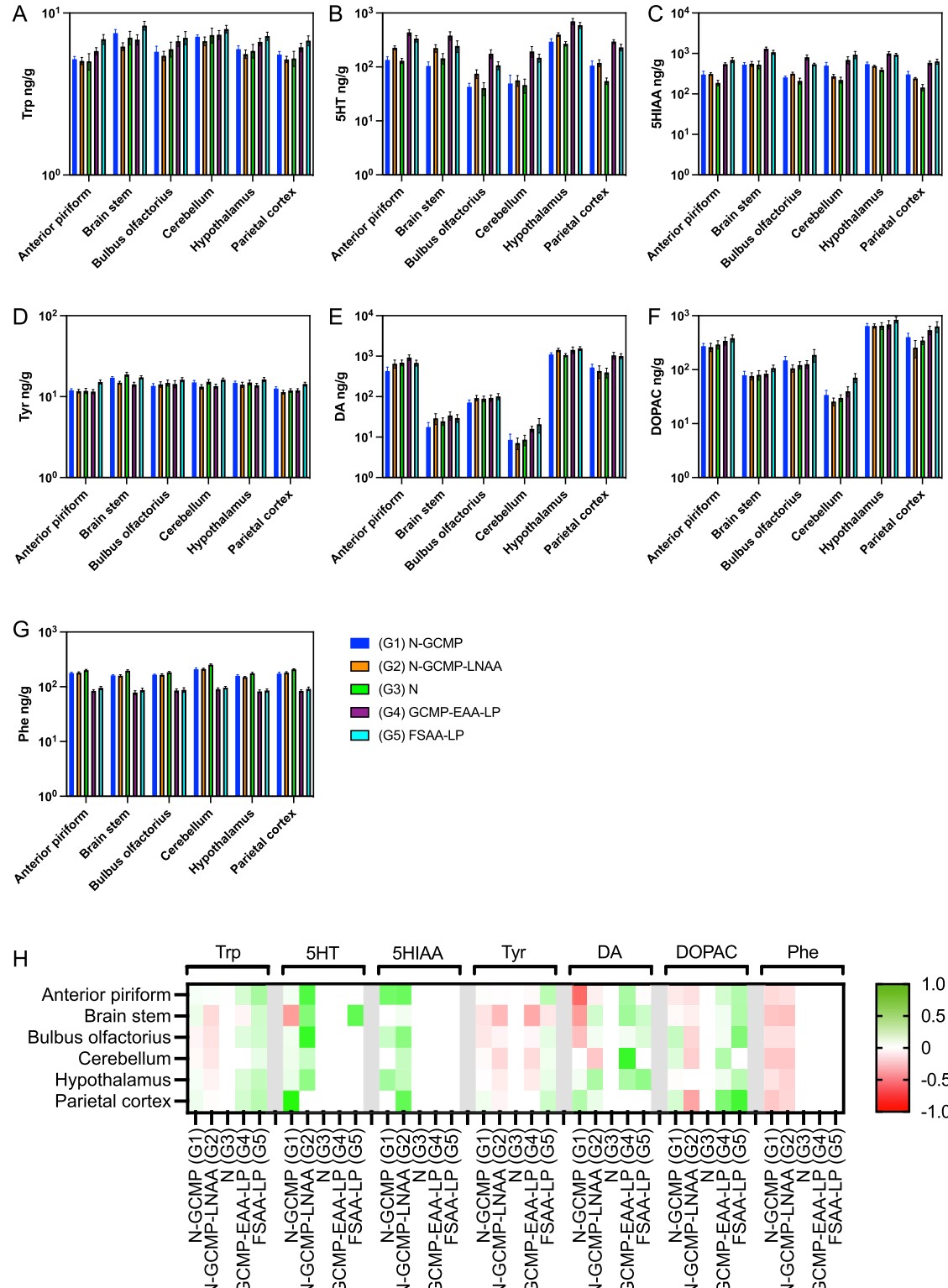

**Fig 6. The content of DA, 5-HT, 5-HIAA, DOPAC, Phe, Tyr and Trp in the 5 groups of animals measured separately in the different brain parts; cerebellum, brain stem, hypothalamus, parietal cortex, anterior piriform, cortex and olfactory bulb.** Trp is metabolized to 5-HT in the brain and finally further metabolized mainly to 5-HIAA. Tyr is precursor of DA which is further converted to DOPAC.

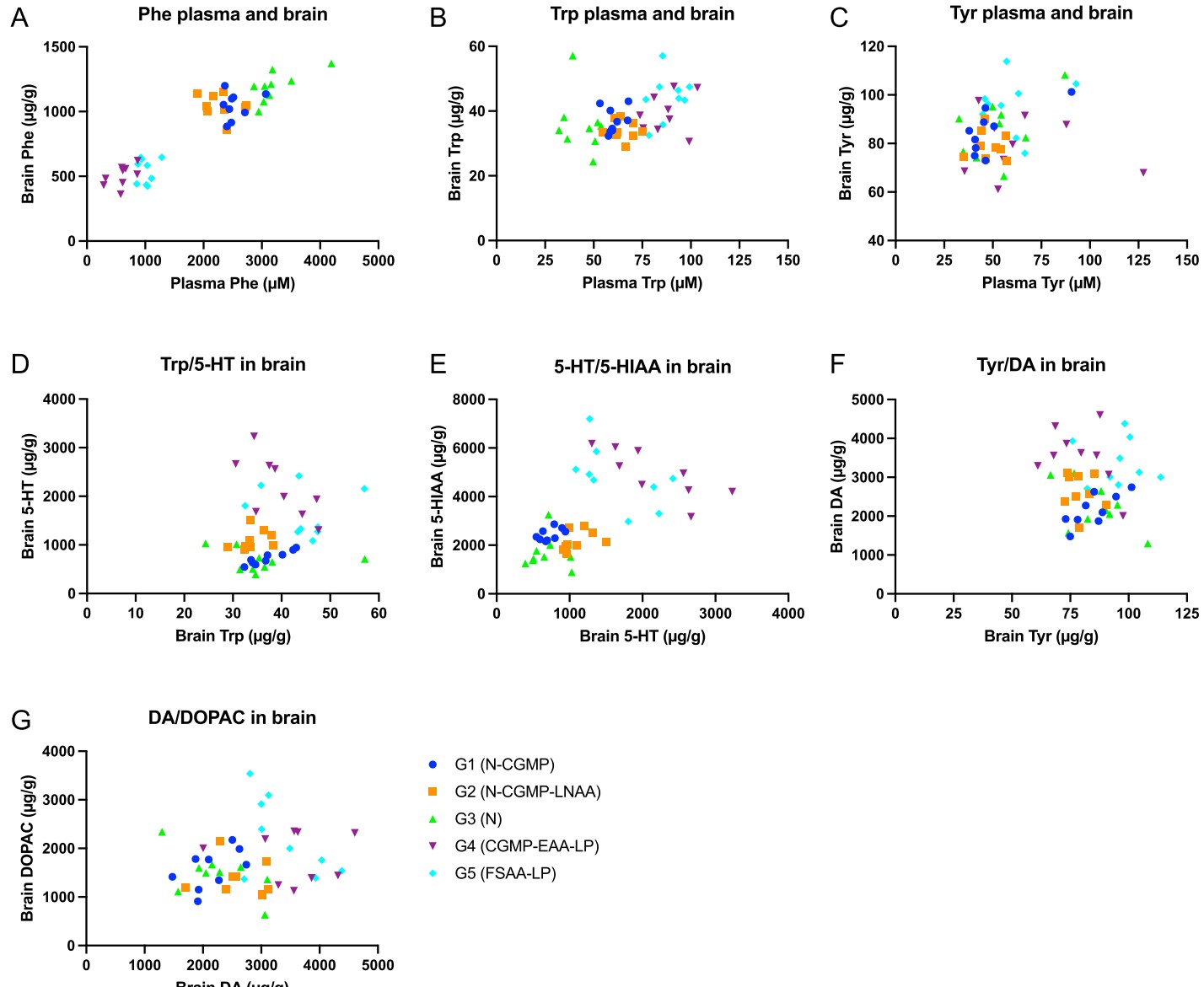

**Fig 7. Correlation of the content of Phe, Tyr, Trp (plasma and brain) and the neurotransmitters, 5-HT, DA, and their metabolites 5-HIAA, DOPAC in the total brain.** Group 1–5 represent: G1(N-CGMP), G2 (N-CGMP-LNAA), G3 (N), G4 (CGMP-EAA-LP) and G5 (FSAA-LP).

scissors was used to cut the individuals with long incisors. It occurred in all groups and varied from 21–40 times per group (p = 0.38), or 0–12 times per mouse during the 12-week test period. Total intake of food was not related to number of teeth cuts. The highest number of cuts occurred in G4 (CGMP-EAA-LP) (40 times), followed by G2 (N-CGMP-LNAA), (27 times), G3 (N) (26 times), G1(N-CGMP), (21 times) and G5 (FSAA-LP), (14 times) with the lowest number of cuts.

## Discussion

Performing study 1, the first question we wanted to clarify was if pure CGMP or CGMP supplemented with the LNAAs given off diet are able to lower the content of Phe in the brain. As

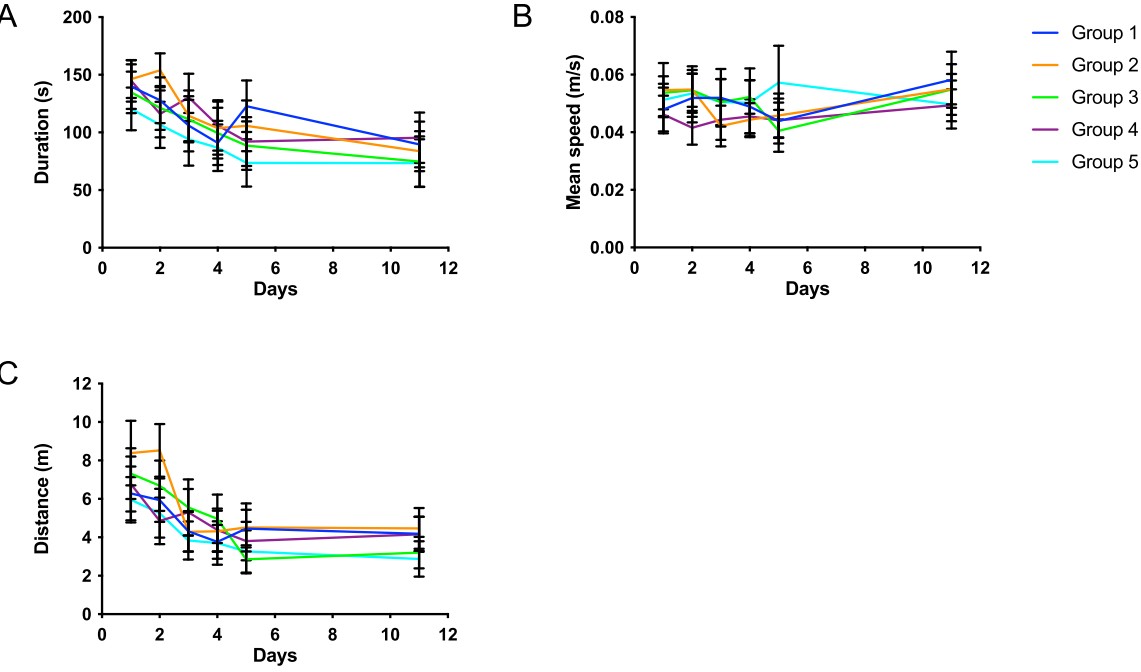

**Fig 8. Barnes maze.** A: Progression of test mean duration before the task completion (maximum duration 180 sec) with error bars indicating standard deviation, significant differences from G3 were found for the following: G2, day 1(p = 0.009) and G1, day 4 (p = 0.036). B: Test mean speeds of animals, (±SD); G3 had significantly higher speed at day 1 compared to G4, (p = 0.002) and significantly slower speed at day 4 compared to G5, (p = 0.026). C: Mean distance (±SD) travelled by the animals during testing; G2 had significantly longer distance at day 1 compared to G3 (p = 0.049), and G3 had significant longer distance, at day 4 compared to G1 (p = 0.010), and G2 (p = 0.019). Group 1–5 represent G1(N-CGMP), G2 (N-CGMP-LNAA), G3 (N), G4 (CGMP-EAA-LP) and G5 (FSAA-LP).

both G1 (N-CGMP) and G2 (N-CGMP-LNAA), had significantly lower plasma and brain Phe concentration compared to G3 (N), at week 12. This study confirmed that mice off diet supplemented with CGMP had significantly lower Phe in plasma and total brain compared to a control group fed a normal diet. This is important, since many PKU patients worldwide are off diet, but could benefit from CGMP treatment.

Other studies conducted in wild type and PKU mice fed diets containing either 20% protein from casein, Phe-free synthetic AA or CGMP supplemented with limiting indispensable AA have revealed promising results. In one study, concentrations of Phe in the plasma decreased by 11% and in 5 regions of the brain by 20% when compared to Phe-free synthetic AA [12].

The second question we wanted to clarify was if the supplementation of extra LNAA results in significantly lower brain Phe level compared to supplementation of CGMP alone. At week 6 the Phe concentration was significantly lower in G2 (N-CGMP-LNAA) (but not in G1 (N-CGMP)) compared to G3 (N), but the only significant difference between G1 (N-CGMP) and G2 (N-CGMP-LNAA) was obtained at week 9, where the plasma Phe concentration was significantly higher in G1 (N-CGMP) compared to G2 (N-CGMP-LNAA). Based on these results we might conclude that the additional LNAA did not have any big impact on the brain Phe concentration.

A large number of studies have confirmed, that LNAA have the ability to reduce Phe entering the brain [18, 19, 38–41]. As specified in the beginning, the Phe blocking effect between blood and brain were relevant to investigate for G1(N-CGMP) and G2 (N-CGMP-LNAA), since these diets had very similar composition, except that LNAA was added solely to G2

(N-CGMP-LNAA). The concentration of plasma Trp increased significantly in both G1 (N-CGMP) and G2 (N-CGMP-LNAA), from week 6 to week 12, and both groups showed similar levels of Tyr and Trp in both plasma and brain. No increase in concentration of plasma Trp was observed for G3 (N). Since CGMP already contain large amounts of Thr, Val and Ile, these AAs could potentially be sufficient to block passage of Phe across the BBB. However as the Phe concentration in det given to G3(N) contains more Phe compared to both G1 (N-CGMP) and G2(N-CGMP-LNAA), (1.01 versus 0.74 g/100g; approximately 36% more) this difference could also account for the increased plasma Phe concentration at week 12 (3253 versus 2533 and 2296 µM/l, approximately 28–41% more).

However, some benefit from adding LNAA to the diet was observed according to the growth, bone density and brain serotonin (5-HT). At the end of the study, G2 (N-CGMP-LNAA) had a significantly higher weight compared to both G1 (N-CGMP) and G3 (N) and furthermore a significant increased aBMD was observed for G2 (N-CGMP-LNAA) compared to the control group, G3 (N).

The aim according to study 2 was to address the following question: Does a combination of CGMP, EAAs and LP diet, provide similar plasma and brain Phe levels, growth and behavioral skills as a formula with a similar combination of pure FSAA.

Both the plasma and the brain concentration of Phe in G4 CGMP-EAA-LP) and G5 (FSAA-LP) were significantly reduced compared to G3 (N). Furthermore, the diet containing CGMP, EAAs and LP diet seems more potent in reducing the plasma Phe concentration compared to the formula containing FSAA as the concentration of Phe in the plasma in G4 CGMP-EAA-LP) was significantly reduced compared to the concentration in G5 (FSAA-LP). However, this effect could be related to the higher intake of phe-containing water (26% more in G5(FSAA-LP), compared to G4 (CGMP-EAA-LP).

Investigation of the growth revaled that both formula leads to higher weight, BMC, aBMC and femur lengt compared to normal diet (N). Again the formula G4 (CGMP-EAA-LP) seems slighly more potent as the highest average growth compared to start was obtain for G4 (CGMP-EAA-LP), compared to G5 (FSAA-LP).

Barnes Maze test has been successfully used in other mice studies [42, 43]. However, this test may not have been sensitive enough for this specific mouse strain, since it was (with a few exceptions) not possible to find any significant differences in behavior. The strain has been used in other studies at Aarhus University [44] and *Bruinenberg et all* found that the genetic background of the mice could play an important role [45]. An explanation could be, that the mice did not start the diet intervention before week 4, as diet intervention before this time is very difficult. A study has shown that week 3 is a very critical period and could potentially already have affected the brain [22]. Other explanations for this could be that the number of animals per group were too low and/or the test period too short to show a difference. Since the main objective of the study was to determine the differences between the diet compositions, we did not include a wild type mouse. However, this could have been useful for the maze study, since there were no differences between the groups, despite the different diet regimes. Similar studies have used wild type mouse to compare with [12, 46]

Thus al together we can conclude that a combination of CGMP, EAAs and LP diet, provide similar plasma and brain Phe levels, growth as a formula with a similar combination of pure FSAA. According to behavioral skills, we are unfortunately not able to make any conclusion.

It is noteworthy that several AA (Leu, Ile, Met, Val, His and Arg) increased significantly for both G4(CGMP-EAA-LP) and G5(FSAA-LP) from week 6 to week 12, and also that Ile, Leu and Val demonstrated significantly higher values for G4(CGMP-EAA-LP) compared to G5 (FSAA-LP) after week 12. This indicate that the higher level of Ile and Val reflect a better absorption due to the natural content of these specific AAs in CGMP.

The effect of the individual diets in G4(CGMP-EAA-LP) and G5 (FSAA-LP) on Phe levels are recognizable during the entire period, already after the first week of treatment. Since both G1(N-CGMP) and G2 (CGMP-LNAA-A) were off diet, we expected that Phe would be higher in these groups compared to G4 (CGMP-EAA-LP) and G5 (FSAA-LP).

The high Phe concentration in G3(N) was expected due to the casein diet. Phe increased significantly from week 6 to week 12 in G3 (N), which is in line with studies showing, that Phe builds up in plasma over time and results in irreversible brain damage. Our result demonstrate that the mice fed on normal diet (G3) have shorter femur length, lower BNC and lower aBMD indication that the high Phe level also might affect the bone formation negatively.

## Conclusion

This study provides important data about CGMP supplementation as a possible alternative treatment strategy for PKU. This study verified that it was possible to lower brain Phe significantly with supplementation of CGMP to a semi free (normal) diet. Also, it was confirmed that CGMP in combination with FSAA as supplement to LP diet had the same effect on plasma- and brain Phe levels, growth as a formula with a similar combination of FSAA. Treatment of PKU is already successful, but new alternatives are necessary in order to improve compliance [47]. CGMP can provide a safe and efficient supplement to a semi free- as well as a LP diet as an alternative to FSAA products. Further long-term studies in humans are needed to support the findings from the present study.

## Acknowledgments

The authors express their gratitude to Dorte Hermansen, Benedicte Vestergaard Jensen and Jani Kær for assistance in the animal facility.

## Author Contributions

**Conceptualization:** Kirsten K. Ahring, Frederik Dagnæs-Hansen, Annemarie Brüel, Erik Jensen, Thomas G. Jensen, Karen Brøndum-Nielsen, Michael Pedersen, Mads Kjolby, Lisbeth B. Møller.

**Data curation:** Kirsten K. Ahring, Annemarie Brüel, Mogens Johannsen, Karen S. Johansen, Mads Kjolby.

**Formal analysis:** Kirsten K. Ahring, Annemarie Brüel, Mette Christensen, Erik Jensen, Mogens Johannsen, Michael Pedersen, Lambert K. Sørensen, Mads Kjolby, Lisbeth B. Møller.

**Funding acquisition:** Kirsten K. Ahring, Erik Jensen.

**Investigation:** Kirsten K. Ahring, Frederik Dagnæs-Hansen, Karen S. Johansen, Jesper G. Madsen, Michael Pedersen, Lambert K. Sørensen, Mads Kjolby.

**Methodology:** Kirsten K. Ahring, Mette Christensen, Mogens Johannsen, Michael Pedersen, Lambert K. Sørensen, Mads Kjolby, Lisbeth B. Møller.

**Project administration:** Kirsten K. Ahring, Frederik Dagnæs-Hansen, Erik Jensen, Allan M. Lund, Lisbeth B. Møller.

**Resources:** Thomas G. Jensen.

**Software:** Mads Kjolby.

**Supervision:** Frederik Dagnæs-Hansen, Annemarie Brüel, Erik Jensen, Mogens Johannsen, Karen S. Johansen, Allan M. Lund, Karen Brøndum-Nielsen, Michael Pedersen, Lambert K. Sørensen, Mads Kjolby, Lisbeth B. Møller.

**Validation:** Mette Christensen, Lambert K. Sørensen.

**Writing – original draft:** Kirsten K. Ahring.

**Writing – review & editing:** Frederik Dagnæs-Hansen, Annemarie Brüel, Mette Christensen, Erik Jensen, Thomas G. Jensen, Mogens Johannsen, Karen S. Johansen, Allan M. Lund, Jesper G. Madsen, Karen Brøndum-Nielsen, Michael Pedersen, Lambert K. Sørensen, Mads Kjolby, Lisbeth B. Møller.

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
