## [Decision Letter · Decision Letter 0]

26 Feb 2020

PONE-D-19-32961

Effect of offering casein glycomacropeptide versus free synthetic amino acids to early treated phenylketonuria mice

PLOS ONE

Dear Dr. Ahring,

Thank you for submitting your manuscript to PLOS ONE. After careful consideration, we feel that it has merit but does not fully meet PLOS ONE’s publication criteria as it currently stands. Therefore, we invite you to submit a revised version of the manuscript that addresses the points raised during the review process.

Both reviewers raised some comprehensible matters, which should improve the quality of your paper and render it finally acceptable. Please seriously consider and respond to all these points one by one.

In particular, data from untreated controls would be appreciated for comparison, which can also be via reference to another paper containing this information. With regard to statistics reviewer 1 suggests one-way ANOVA instead of multiple t tests - this could be more convioncing, but multiple t tests are formally correct, so I am inclined to leave the final choice on your side.

Here follows an additional comment from an expert reviewer, which appears meaningful: *"**There is a long history in the field that simply taking large amounts of large neutral amino acids orally can help protect the brain from damage from elevated phenylalanine. This is all based upon an overly simplistic model of amino acid transport at the blood brain barrier. Investigators continue to work to prove this model even though their own data disprove it and they keep doing statistical manipulations to try to 'prove' their point instead of just taking the data at face value and accepting what the data are showing them which is that in the face of very elevated blood phenylalanine, no amount of amino acid supplementation will really effect brain phenylalanine unless it accomplishes the goal of substantially also lowering the blood phenylalanine."* You should at least discuss this critical view about the present state of research in the discussion section.

We would appreciate receiving your revised manuscript by Apr 11 2020 11:59PM. To enhance the reproducibility of your results, we recommend that if applicable you deposit your laboratory protocols in protocols.io, where a protocol can be assigned its own identifier (DOI) such that it can be cited independently in the future. For instructions see: http://journals.plos.org/plosone/s/submission-guidelines#loc-laboratory-protocols

We look forward to receiving your revised manuscript.

Kind regards,

Clemens Fürnsinn, Ph.D.

Academic Editor

PLOS ONE

Journal Requirements:

"KKA was an industrial PhD student sponsored by Arlafoodsingredients (AFI) and The Danish Agency for Science, Technology and Innovation and this manuscript is part of the Ph.D. thesis. EJ is employed as food scientist at AFI and contributed with his knowledge about the product GMP."

We note that one or more of the authors have an affiliation to the commercial funders of this research study : 'Arla Foods Ingredients Group P/S, Viby J, Denmark'.

Reviewers' comments:

Reviewer's Responses to Questions

**Comments to the Author**

1. Is the manuscript technically sound, and do the data support the conclusions?

Reviewer #1: Partly

Reviewer #2: Yes

2. Has the statistical analysis been performed appropriately and rigorously? 

Reviewer #1: No

Reviewer #2: Yes

3. Have the authors made all data underlying the findings in their manuscript fully available?

Reviewer #1: Yes

Reviewer #2: Yes

4. Is the manuscript presented in an intelligible fashion and written in standard English?

Reviewer #1: Yes

Reviewer #2: Yes

5. Review Comments to the Author

Reviewer #1: The authors report on experiments to compare the effect of adding casein glycomacropeptide (CGMP) to a casein based diet upon the hyperphenylalaninemia phenotype in Pahenu2 mice, basically in an attempt to see if GMP might be work similarly to large neutral amino acid (LNAA) supplements in untreated PKU. They compared the effects to mice treated with dietary protein restriction with or without CGMP supplement. The manuscript is generally well written but there are several typographic errors including a recurring one that says ‘Error! Reference source not found.’

The clarity of the statistical analysis used in the study needs to be improved. Vast numbers of p values are provided, primarily in the supporting information, between many variables, but how these p values were generated is not noted. The statistics sections states the use of t tests and ANOVAs but the results sections never state when these are used. From my reading, five different treatment groups in all female mice were established from the outset. The only viable statistical method then would be a one-way ANOVA to examine the overall effect of treatment across the five groups with a post hoc intergroup comparison thereafter. There would be no place for choosing to employ t tests between multiple pairs of groups as this would lead to compounded errors.

By inspection of the final blood Phe concentrations, I would estimate, if ANOVA were used, a significant treatment effect across all groups given the large difference between the groups on casein and those on Phe restriction; however, it is difficult to estimate whether intergroup differences reach statistical significance between the animals on casein but receiving CGMP or not. More importantly, I doubt intergroup differences in brain Phe, Tyr, or Trp or any of the neurotransmitters are significant at all in the groups that received a casein containing diet.

I recommend that the bar graphs be converted to whisker plots so that the true mean and range of the data can be inspected. This would apply to Figure 3 and supplementary Figure 1.

For Figure 3, I recommend that the amino acids be separated from the neurotransmitters into different charts as the amounts measured differ by a two orders of magnitude and it makes inspection of the current figure difficult.

If it’s possible to find a suitable short phrase or abbreviation, I recommend naming the actual treatment of the groups in the figures rather than using Group 1, group 2, etc in the figures as I found myself having to go back repeatedly to refresh my memory on what each treatment group actually was. I can remember a couple treatments but not five.

As the authors have clearly learned, the lack of either wild type B6 mice or untreated B6-Pahenu2 mice as controls in the behavioral studies make the current results difficult to interpret. Given the decrease in distance traveled, the animals are clearly learning the maze, which I suspect is an improvement over what they would have done had they not been treated from early in life, but do the authors have any maze data on wild type mice or untreated B6-Pahenu2 mice (not collected contemporaneously obviously) to compare to?

The timing of the dietary treatment is bit unclear from the manuscript. It is stated in the manuscript that the pups were products of homozygous Pahenu2 dams treated with Phe-restricted diet and this diet was continued through weaning. Figure 1 shows experimental diets being initiated sometime between week 4-16 and continuing to week 19. When precisely did the experimental diet start? What diet were the mice fed between weaning and the onset of the experimental diet. This would have influenced their ability to perform the maze testing.

Incidentally, the use of the low Phe diet in the dams clearly allows them to generate progeny but since their milk would contain normal concentrations of lactalbumin, I would except Pahenu2 homozygous pups to become hyperphenylalaninemic regardless of whether the dam continued on a low Phe diet or not. Do the authors have blood Phe data on the progeny at weaning or prior to the institution of the experimental diets? The pups may have suffered sufficient brain damage from hyperphenylalaninemia during the juvenile period that Phe lowering treatment instituted later in life may have had little effect on behavior in the animals.

The current concept of LNAA transport at the blood brain barrier being mediated solely via the LAT-1 transporter is exceedingly inadequate. There is evidence for a number of other transporters, some that transport amino acids in reverse direction against the gradient, being involved in brain amino acid homeostasis. The existence of this system is why dietary manipulation in this experiment had little effect upon brain amino acid content other than Phe.

The authors make the statement in their introduction that the imbalance in brain LNAA is ‘probably the primary cause of disrupted brain development in this disorder’ and then cite a single reference. This statement denies abundant evidence and dozens of other publications on a multitude of other potential pathogenic mechanisms; the statement should be eliminated.

The authors find improved bone density primarily in the groups of animals on the Phe-restricted diets yet make no statement about the potential pathogenesis of hyperphenylalaninemia itself upon bone health.

The authors conclude that CGMP can be a ‘relevant supplement for the treatment of PKU.’ I don’t disagree. However, in my opinion, the more important conclusion is that dietary Phe-restriction, potentially using CGMP as part of the Phe-restricted diet, leads to vastly superior outcomes using multiple measures in comparison to ignoring Phe restriction and trying to supplement with LNAA.

Reviewer #2: This is a valuable study, but the separation of Phe intake through water from all other AA in G4 and G5 makes it difficult to compare data. The text is very dry to read and I would recommend to combine results and discussion to give the reader more guidance. Some of the discussion is just a repeat of results. Please also include the supplementary data in the main part of the manuscript.

Page 13 diets: Please explain diet composition of G3.

Page 13 of submitted pdf; Phenylalanine supplementation: 57mg/ml would be an extraordinary amount of Phe (345 mM) and not suitable for the mouse strain. Probably 57mg/L.

Page 19/20: There are two error messages in the text.

Table 2: Threonine is elevated in G4 and G5. It is an essential AA and diet 5 has a significant amount, but diet 4 doesn't seem to have Thr. Please explain. Arginine is consistent and found in G4 and G5 diets.

I would recommend to convert table 2 into a bar graph.

Figures in the pdf are of unacceptable resolution.

Fig 3 Please present control data from a background mouse strain. The figure caption needs to explain the information content of the right panel. Do not abbreviate brain areas.

6. PLOS authors have the option to publish the peer review history of their article (what does this mean?). If published, this will include your full peer review and any attached files.

Reviewer #1: Yes: Cary O. Harding

Reviewer #2: No

---

## [Author Response · Author response to Decision Letter 0]

27 Sep 2020

Once again, we thank you for the time you put in reviewing our paper and look forward to meeting your expectations. Since your inputs have been precious, in the eventuality of a publication, we would like to acknowledge your contribution explicitly.

---

## [Decision Letter · Decision Letter 1]

8 Oct 2020

PONE-D-19-32961R1

Effect of offering casein glycomacropeptide versus free synthetic amino acids to early treated phenylketonuria mice

PLOS ONE

Dear Dr. Ahring,

Thank you for submitting your manuscript to PLOS ONE. After careful consideration, we have decided that your manuscript does not meet our criteria for publication and must therefore be rejected.

It was the reviewer´s feeling that the extent of changes and re-writing of the manuscript was not sufficient (and that there was not a substantiated rebuttal outlining why this was so). I must say that I share the impression that several points remained unclarified and that the changes were less extensive than suggested by the reviewers. Your response to the reviewers is rather brief and falls back behind the point-by-point explanations that we are used to receive with detailed elaboration and reasoning of which changes have been made (or not made), and why.

I am sorry that we cannot be more positive on this occasion, but hope that you appreciate the reasons for this decision.

Yours sincerely,

Clemens Fürnsinn, Ph.D.

Academic Editor

PLOS ONE

Reviewers' comments:

Reviewer's Responses to Questions

**Comments to the Author**

1. If the authors have adequately addressed your comments raised in a previous round of review and you feel that this manuscript is now acceptable for publication, you may indicate that here to bypass the “Comments to the Author” section, enter your conflict of interest statement in the “Confidential to Editor” section, and submit your "Accept" recommendation.

Reviewer #2: (No Response)

2. Is the manuscript technically sound, and do the data support the conclusions?

Reviewer #2: Yes

3. Has the statistical analysis been performed appropriately and rigorously? 

Reviewer #2: Yes

4. Have the authors made all data underlying the findings in their manuscript fully available?

Reviewer #2: Yes

5. Is the manuscript presented in an intelligible fashion and written in standard English?

Reviewer #2: No

6. Review Comments to the Author

Reviewer #2: The study requires a significant rewrite. Results and Discussion should be combined. The concentration of phenylalanine given to the animals does not have correct units.

7. PLOS authors have the option to publish the peer review history of their article (what does this mean?). If published, this will include your full peer review and any attached files.

Reviewer #2: No

- - - - -

---

## [Author Response · Author response to Decision Letter 1]

30 Apr 2021

To whom it may concern

My colleagues and I humbly re-submit our manuscript entitled “The effect of Casein glycomacropeptide versus free synthetic amino acids for early treatment of phenylketonuria in a mice model” for your re-consideration. 

We wish to thank you all for your constructive comments in this second round of review. Your comments provided valuable insights to refine its contents and analysis. In this document, we try to address the issues raised as best as possible

Authors respond to reviewers PLOS ONE answers

(Reviewer #1) Thank you very much for your suggestion regarding using one-way ANOVA instead of multiple t tests. During my PhD study period, I did seek advice at the Statistical Department at the University of Copenhagen and the teacher recommended that I should use multiple t-tests in order to obtain the most transparent results. He stated, that in his opinion it can confuse the outcome to include to many parameters in the same statistical analysis. Therefore, I choose to keep this method as kindly offered from the Academic Editor. 

I recommend that the bar graphs be converted to whisker plots so that the true mean and range of the data can be inspected. This would apply to Figure 3 and supplementary Figure 1.

Thank you for the suggestion, we have converted the bar graphs Figure 3 and supplementary Figure 1 to whisker plots.

For Figure 3, I recommend that the amino acids be separated from the neurotransmitters into different charts as the amounts measured differ by two orders of magnitude and it makes inspection of the current figure difficult.

Thank you for the recommendation for Figure 3, we have separated the amino acids from the neurotransmitters into different charts to make the inspection of the current figure easier.

If it’s possible to find a suitable short phrase or abbreviation, I recommend naming the actual treatment of the groups in the figures rather than using Group 1, group 2, etc in the figures as I found myself having to go back repeatedly to refresh my memory on what each treatment group actually was. I can remember a couple treatments but not five.

We do agree that a suitable short phrase or abbreviation would be preferable for naming the actual treatment of the groups. We have done as follows: Re-named the 5 groups and added this information to glossary.

As the authors have clearly learned, the lack of either wild type B6 mice or untreated B6-Pahenu2 mice as controls in the behavioral studies make the current results difficult to interpret. Given the decrease in distance traveled, the animals are clearly learning the maze, which I suspect is an improvement over what they would have done had they not been treated from early in life, but do the authors have any maze data on wild type mice or untreated B6-Pahenu2 mice (not collected contemporaneously obviously) to compare to?

Thank you for addressing this very important issue regarding lack of wild type mice. We did not have a wild type mouse to compare to, which we initial found was not a problem, since the main objective of the study was to determine the differences between the diet compositions. We are aware that similar studies has used wild type mouse to compare with ( (Ney, 2008) (Vliet, 2016) and we have realized afterwards, that this could have been useful for the Barnes maze study, since there were no differences between the groups, despite the different diet regimes. We have included a comment about this in the discussion section.

The timing of the dietary treatment is bit unclear from the manuscript. It is stated in the manuscript that the pups were products of homozygous Pahenu2 dams treated with Phe-restricted diet and this diet was continued through weaning. Figure 1 shows experimental diets being initiated sometime between week 4-16 and continuing to week 19. When precisely did the experimental diet start? What diet were the mice fed between weaning and the onset of the experimental diet. This would have influenced their ability to perform the maze testing.

The experimental diet started at week 4, right after weaning was ended. The breeding animals were maintained on Phe-free semi synthetic diet (Harlan Laboratories), also during pregnancy and weaning period, a Maternal PKU diet. As the diet was free of Phe, the drinking water was supplemented with Phe (Sigma-Aldrich Chemie) to a final concentration of 62.5 mg/ml during that period.

We have now included a new figure 1 to make it clearer.

Incidentally, the use of the low Phe diet in the dams clearly allows them to generate progeny but since their milk would contain normal concentrations of lactalbumin, I would except Pahenu2 homozygous pups to become hyperphenylalaninemic regardless of whether the dam continued on a low Phe diet or not. Do the authors have blood Phe data on the progeny at weaning or prior to the institution of the experimental diets? The pups may have suffered sufficient brain damage from hyperphenylalaninemia during the juvenile period that Phe lowering treatment instituted later in life may have had little effect on behavior in the animals.

Yes we agree that this could be a problem however It was not possible to take a blood test for determination of the total amino acid profile at week 1 of diet intervention since the mice at that time, were too small to tolerate blood sampling, but as all animals were treated equal before week 1, we assume that the start value was similar in all groups.

The current concept of LNAA transport at the blood brain barrier being mediated solely via the LAT-1 transporter is exceedingly inadequate. There is evidence for a number of other transporters, some that transport amino acids in reverse direction against the gradient, being involved in brain amino acid homeostasis. The existence of this system is why dietary manipulation in this experiment had little effect upon brain amino acid content other than Phe.

The authors make the statement in their introduction that the imbalance in brain LNAA is ‘probably the primary cause of disrupted brain development in this disorder’ and then cite a single reference. This statement denies abundant evidence and dozens of other publications on a multitude of other potential pathogenic mechanisms; the statement should be eliminated

We have removed the sentence as requested by reviewer.

The authors find improved bone density primarily in the groups of animals on the Phe-restricted diets yet make no statement about the potential pathogenesis of hyperphenylalaninemia itself upon bone health.

Thank you for pointing out that statement about the potential pathogenesis of hyperphenylalaninemia upon bone health was missing. We have in the discussion included the impact on bone density.

The authors conclude that CGMP can be a ‘relevant supplement for the treatment of PKU.’ I don’t disagree. However, in my opinion, the more important conclusion is that dietary Phe-restriction, potentially using CGMP as part of the Phe-restricted diet, leads to vastly superior outcomes using multiple measures in comparison to ignore Phe restriction and trying to supplement with LNAA

Also thank you for emphasizing, that it is important to highlight the fact, that taking large amounts of large neutral amino acids orally can help protect the brain from damage from elevated phenylalanine, but the main lowering effect on brain phenylalanine comes from lowering the blood phenylalanine. We thought we already indicated that but have now expanded the discussion section where we describe in more detail with the work of others, as you advised us to do.

Reviewer 2#

This is a valuable study, but the separation of Phe intake through water from all other AA in G4 and G5 makes it difficult to compare data. The text is very dry to read and I would recommend to combine results and discussion to give the reader more guidance. Some of the discussion is just a repeat of results. Please also include the supplementary data in the main part of the manuscript.

Thank you for your valuable advices regarding design of the manuscript. We do agree that the text can be dry to read, and therefore we have combined “results-“and “discussion sections” a bit more as suggested and included the supplementary data in the main part of the manuscript.

Page 13 diets: Please explain diet composition of G3.

The composition of G3 is a complete diet, consisting of casein (MIPRODAN® 30), sucrose, corn starch, corn oil, cellulose, mineral- and vitamin-mix. Basically, the nutritional value is the same as the other experimental groups. We have now included a detailed description of G3 in Table 2

Page 13 of submitted pdf; Phenylalanine supplementation: 57mg/ml would be an extraordinary amount of Phe (345 mM) and not suitable for the mouse strain. Probably 57mg/L.

Thank you for pointing out that 57mg/ml of Phenylalanine supplementation would be an extraordinary amount of Phe (345 mM) and not suitable for the mouse strain. In fact, 25 mg/ml was used to produce the drinking mixture, where 9 ml (225 mg) was added to 391 ml of water, which provided a final concentration of 563 mg/l

Page 19/20: There are two error messages in the text.

Thank you for making us aware of this, it is corrected now.

Table 2: Threonine is elevated in G4 and G5. It is an essential AA and diet 5 has a significant amount, but diet 4 doesn't seem to have Thr. Please explain. Arginine is consistent and found in G4 and G5 diets.

We are glad that you pointed this out, since it is not clear at all from the table 2, that the CGMP product contains large amount of Thr. We have now added the data from CGMP-20 to make it more obvious

I would recommend to convert table 2 into a bar graph. Figures in the pdf are of unacceptable resolution.

We hope the resolution of the figures have improved with the new system: uploading figure files to the Preflight Analysis and Conversion Engine (PACE) digital diagnostic tool https://pacev2.apexcovantage.com/

Fig 3 Please present control data from a background mouse strain. The figure caption needs to explain the information content of the right panel. 

Unfortunately, the mouse strain we used from AAU had no wild type as described in the introduction section, but we have found useful information from other similar studies (Ney, 2008) (Vliet, 2016)

We were aware from the start of the study, that a wild type of the mouse strain would have been a useful tool. However, the information we needed from the study would be possible to gain, also without a wild type, but that fact determined the design of the study: 2 groups with pure GCMP, 1 group on normal diet, 2 groups with FSAA

We expected to find a difference between the groups due to the fact, that late-diagnosed/untreated PKU patients over the years have been studied and observed.

Do not abbreviate brain areas.

We have now changes to full names for brain areas

---

## [Decision Letter · Decision Letter 2]

25 Nov 2021

The effect of Casein glycomacropeptide versus free synthetic amino acids for early treatment of phenylketonuria in a mice model

PONE-D-19-32961R2

Dear Dr. ahring,

We’re pleased to inform you that your manuscript has been judged scientifically suitable for publication and will be formally accepted for publication once it meets all outstanding technical requirements.

Kind regards,

Aneta Agnieszka Koronowicz, PhD

Academic Editor

PLOS ONE

Additional Editor Comments (optional):

The Authors introduce discussion elements into the results section and vice versa, so it is recommended to combine the results and the discussion (Results and Discussion).

Reviewers' comments:

Reviewer's Responses to Questions

**Comments to the Author**

1. If the authors have adequately addressed your comments raised in a previous round of review and you feel that this manuscript is now acceptable for publication, you may indicate that here to bypass the “Comments to the Author” section, enter your conflict of interest statement in the “Confidential to Editor” section, and submit your "Accept" recommendation.

Reviewer #2: All comments have been addressed

2. Is the manuscript technically sound, and do the data support the conclusions?

Reviewer #2: Yes

3. Has the statistical analysis been performed appropriately and rigorously? 

Reviewer #2: Yes

4. Have the authors made all data underlying the findings in their manuscript fully available?

Reviewer #2: Yes

5. Is the manuscript presented in an intelligible fashion and written in standard English?

Reviewer #2: Yes

6. Review Comments to the Author

Reviewer #2: The concerns raised by this reviewer were addressed. I still would recommend to combine results and discussion.

7. PLOS authors have the option to publish the peer review history of their article (what does this mean?). If published, this will include your full peer review and any attached files.

Reviewer #2: No

---

## [Editor Report · Acceptance letter]

7 Dec 2021

PONE-D-19-32961R2 

The effect of Casein glycomacropeptide versus free synthetic amino acids for early treatment of phenylketonuria in a mice model 

Dear Dr. Ahring:

I'm pleased to inform you that your manuscript has been deemed suitable for publication in PLOS ONE. Congratulations! Your manuscript is now with our production department. 

Kind regards, 

on behalf of

Prof. Aneta Agnieszka Koronowicz 

Academic Editor

PLOS ONE